# FCLoRA: Low-Rank Adaptation with Fine-Grained Component Injection

## Abstract

In recent years, low-rank adaptation (LoRA) has emerged as a significant paradigm, which freezes the pre-trained weights and introduces small, learnable adapters instead of fine-tuning the full set of parameters. In this work, we uncover several key insights regarding to the *singular* components of the network parameters based on Singular Value Decomposition (SVD). Firstly, the dominant singular components with large singular values in pre-trained network parameters can be effectively reused during fine-tuning, whereas the fine-grained components with smaller singular values are more task-specific and require substantial adaptation. Secondly, the growth of singular values in the LoRA adapter leads to the forgetting of pre-trained knowledge — a well-known issue called *catastrophic forgetting*. Building upon these observations, we propose **FCLoRA**, which injects learnable fine-grained singular components to the pre-trained model. By employing parameterized SVD and restricting the singular values to an appropriate range, **FCLoRA** can effectively adapt to new tasks by learning in the fine-grained singular domain and alleviates the catastrophic forgetting problem. We conduct extensive experiments and demonstrate that **FCLoRA** not only improves performance but also effectively retains pre-trained knowledge.

## 1 Introduction

Pre-trained language models (PLMs) have achieved remarkable performance in various natural language processing tasks (Devlin et al., 2019; Liu et al., 2019; Lan et al., 2019; He et al., 2020; Touvron et al., 2023a; Achiam et al., 2023; Anil et al., 2023). The common way to adapt pre-trained language models to downstream tasks is *fine-tuning*. However, fine-tuning all parameters and storing copies of the large model for each downstream task results in significant cost and memory consumption. To address this issue, recent studies suggest parameter-efficient fine-tuning (PEFT) methods (Hu et al., 2021; Zhang et al., 2023; Lialin et al., 2023; Liu et al., 2024; Jiang et al., 2024; Meng et al., 2024; Wang et al., 2024), fine-tuning with only a small number of parameters.

Low-Rank Adaptation (LoRA) (Hu et al., 2021), which updates parameters using low-rank matrices, has shown promising performance over other methods such as prompt tuning (Lester et al., 2021) or prefix tuning (Li & Liang, 2021). LoRA keeps the pre-trained weights frozen and updates only a small number of parameters, which makes LoRA both storage- and compute-efficient. LoRA is designed based on the assumption that pre-trained language models are inherently low-dimensional and can learn efficiently even with random projections into smaller subspaces. The low-rank matrices serve as adapters, amplifying features that were learned but not emphasized during pre-training.

In recent years, many studies have investigated the properties of *singular components* with Singular Value Decomposition (SVD) in LoRA (Meng et al., 2024; Wang et al., 2024; Bałazy et al., 2024). A singular component refers to a single rank-1 matrix formed by the product of a pair of left and right singular vectors and their corresponding singular value. Specifically, a *dominant* singular component refers to one associated with a relatively larger singular value, representing the global structure of the matrix (Abdi & Williams, 2010; Meng et al., 2024). Conversely, a *fine-grained* singular component corresponds to a relatively smaller singular value and is often considered as noise (Wang et al., 2024). In deep learning, however, because learned weight matrices are typically full rank (Hu et al., 2021; Garg et al., 2025; Yu & Wu, 2023), the fine-grained singular components are not merely noise; rather, they also encode detailed and fine-grained information within the matrix.

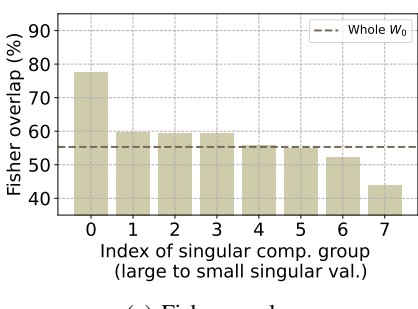

(a) Fisher overlap

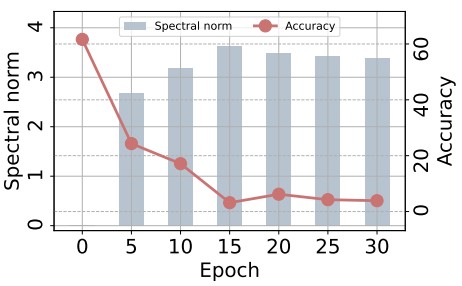

(b) Trade-off between spectral norm and accuracy

Figure 1: (a) The Fisher overlap (Kirkpatrick et al., 2017) between the pre-trained task (Book-Corpus) and the fine-tuning task (MRPC), evaluated over partial reconstructions of the pre-trained network parameters obtained by grouping singular components sorted from large to small according to their singular values. Additional visualizations for other datasets are in Appendix E. (b) The trade-off between the spectral norm of the adapter and the accuracy on the pre-trained task (BookCorpus) during fine-tuning of LoRA on the STS-B dataset from the GLUE benchmark for RoBERTa$_{\text{base}}$.

**Motivations.** We uncover key insights on the *singular components* of the network parameters.

Firstly, the dominant singular components of the pre-trained network parameters can be reused for the fine-tuning task to a great extent; the fine-grained singular components become more task-specific and thus require a significant adaptation. To quantify the alignment between pre-training and fine-tuning tasks, we compute the *Fisher overlap* (Kirkpatrick et al., 2017; Yao & Hansen, 2022; Qian et al., 2024) based on partial reconstructions of the pre-trained network parameter, obtained by grouping its singular components. Specifically, we perform SVD on the pre-trained parameters, sort the singular values in descending order, and divide the corresponding singular components into groups. Each group is then used to independently reconstruct a partial version of the parameters. The Fisher overlap computed from each reconstruction reflects how well that particular subset of singular components from the pre-trained parameters aligns with the fine-tuning task. A higher Fisher overlap indicates stronger alignment, suggesting that the corresponding components are more transferable. The detailed formulation of the Fisher overlap is provided in Appendix E. Fig. 1 (a) shows that overlap gradually decreases as the singular components become more fine-grained, suggesting that the dominant singular components are already aligned, while fine-grained singular components need more task-specific adaptation.

Secondly, we demonstrate that the growth of singular values in the adapters during fine-tuning leads to forgetting of the pre-trained knowledge. The optimization of deep learning, including LoRA, can be seen as a process of performing a maximum a posteriori (MAP) estimation on the training data. During fine-tuning, the MAP objective maximizes the posterior probability by combining the likelihood of the fine-tuning data with the prior distribution from the pre-training data. We reveal that when the singular values of the adapter increase, the prior from the pre-trained task decreases (see Theorem 3.1). This can result in a phenomenon called *catastrophic forgetting*, where the model rapidly forgets the pre-trained knowledge during fine-tuning. This phenomenon undermines the scalability and reliability of pre-trained models, thereby making it essential to address this issue (Wang et al., 2024; Yang et al., 2024b; Ren et al., 2024; Yang et al., 2024a; Dou et al., 2024). It is also known that during typical stochastic optimization, the spectral norm of weight matrices tends to grow rapidly. As a result, the growth of norm leads to catastrophic forgetting in common fine-tuning scenarios. Fig. 1 (b) shows that LoRA experiences a significant increase in the singular values of the adapter during fine-tuning. This increase is associated to performance degradation on the pre-trained task, suggesting that LoRA is also vulnerable to catastrophic forgetting.

**Main idea.** Inspired by these observations, we propose a **Lo**w-**R**ank **A**daptation with **F**ine-grained **C**omponent injection, called **FCLoRA**, which effectively adapts to the new task while retaining the pre-trained knowledge. We propose to inject an appropriate range of fine-grained singular components into the pre-trained model through parameterized SVD. Restricting the singular values of the injected components prevents them from becoming excessively large, allowing the introduced

modules to maintain a focus on the fine-grained information. This approach helps the model to adapt effectively to new tasks by focusing on the singular components that require greater adaptation. Moreover, the model preserves the pre-trained knowledge by balancing the likelihood from fine-tuning data with prior probabilities from the pre-training dataset. We conduct extensive experiments to evaluate the effectiveness of **FCLoRA**, demonstrating that it consistently outperforms LoRA and its variants across various tasks. Additionally, we assess catastrophic forgetting across multiple baseline models, showing that **FCLoRA** significantly mitigates the forgetting of pre-trained knowledge. Our key contributions can be summarized as follows:

- In Section 3.1, we reveal that the fine-grained singular components of the network parameter require a significant adaptation, and the growth of singular values in the adapters leads to the forgetting of pre-trained knowledge.
- In Section 3.2, we propose an advanced low-rank adaptation method, called **FCLoRA**, which injects the models with the fine-grained singular components using parameterized SVD, ensuring that the pre-trained model efficiently adapts to new tasks and mitigates the catastrophic forgetting problem.
- In Section 4 and Section 5, we conduct comprehensive experiments demonstrating that **FCLoRA** efficiently adapts to the new task and discuss how **FCLoRA** differs from existing LoRA variants, particularly in addressing the limitations, e.g., catastrophic forgetting.

## 2 PRELIMINARIES & RELATED WORKS

### 2.1 TRANSFORMERS

Transformers can be understood from two key submodules: multi-head attention (MHA) and feed-forward network (FFN). The MHA with $h$ parallel heads performs the attention function as follows:

$$\text{MHA}(X) = \text{Concat}(\text{head}_1, \ldots, \text{head}_h)W_o; \quad \text{head}_i = \text{Softmax}\left(\frac{XW_{q_i}(XW_{k_i})^\top}{\sqrt{d_k}}\right)XW_{v_i}, \quad (1)$$

where $W_o \in \mathbb{R}^{d \times d}$ is an output projection weight and $W_{q_i}, W_{k_i}, W_{v_i} \in \mathbb{R}^{d \times d_h}$ are query, key, and value projection weights for each head $i$. $d_h$ is typically set to $d/h$. FFN performs two linear transformations with a ReLU activation as follows:

$$\text{FFN}(X) = \text{ReLU}(XW_{f_1} + b_1)W_{f_2} + b_2, \quad (2)$$

where $W_{f_1} \in \mathbb{R}^{d \times d_m}$ and $W_{f_2} \in \mathbb{R}^{d_m \times d}$. These architectures enable a model to understand the language patterns and generate human-like texts in natural language processing.

### 2.2 LOW-RANK ADAPTATION

LoRA (Hu et al., 2021) suggests the low-rank update of the pre-trained weights by the product of two low-rank matrices. For $h = W_0 x$, the modified forward pass becomes:

$$h = W_0 x + \Delta W x = W_0 x + BAx, \quad (3)$$

where $W_0, \Delta W \in \mathbb{R}^{d_1 \times d_2}$, $A \in \mathbb{R}^{r \times d_2}$ and $B \in \mathbb{R}^{d_1 \times r}$ with $r \ll \{d_1, d_2\}$. $A$ is initialized with a random Gaussian initialization and $B$ with zero, so $\Delta W = BA$ is initially zero at the beginning of training. After fine-tuning, the learnable adapter $\Delta W$ can be integrated into the pre-trained weight $W$ without modifying the original model architecture or adding any additional inference overhead.

**LoRA with *explicit* SVD.** Recent studies have explicitly decomposed the network parameters using SVD to initialize adapters with a subset of components. LoRA-XS (Bałazy et al., 2024) directly decomposes the pre-trained networks and initializes the adapters with principal components. PiSSA (Meng et al., 2024) assumes that the principal components hold the most important information, decomposing the network parameters into principal and residual components using explicit SVD. Then the residual components freeze, while the adapter is initialized with the principal components and directly updated. Conversely, MiLoRA (Wang et al., 2024) proposes directly modifying the minor components of the pre-trained networks, assuming they are noisy and less important, in order to better preserve the pre-trained knowledge.

**LoRA with *parameterized* SVD.** AdaLoRA (Zhang et al., 2023) dynamically adjusts the rank for each LoRA layer based on a sensitivity-driven importance score. They focus on pruning the number of *ranks* with parameterized SVD to meet a predefined budget using heuristic importance scores. LoRA$^2$ (Zhang et al., 2024) uses the twice-nested parameterized SVD to iteratively project the token representations onto *mutually orthogonal planes*. Mo-SARA (Gu et al., 2024) initializes the singular vectors with principal components of the pre-trained network parameters. They freeze the singular vectors and fine-tune only the randomly initialized singular values under the *same eigenvector mappings* with the pre-trained network parameters. Therefore, recent studies design LoRA based on the parameterized SVD (i.e., Equation (9)), which differ in their specific design strategies.

## 2.3 CATASTROPHIC FORGETTING AND LoRA

Catastrophic forgetting refers to the phenomenon where the models forget previously acquired knowledge during adaptation to new tasks, a well-known issue in the field of deep learning (McCloskey & Cohen, 1989; French, 1999; Kirkpatrick et al., 2017). To address this challenge, recent studies have proposed various approaches, including knowledge distillation (Li & Hoiem, 2017; Hou et al., 2019), rehearsal (Riemer et al., 2018; Yang et al., 2023) and dynamic architectures (Yan et al., 2021). This issue is particularly severe in large language models (LLMs), which learn extensive world knowledge through the pre-training process on massive datasets. During the fine-tuning process, where task-specific information is learned based on this world knowledge, forgetting the pre-trained knowledge can significantly undermine the stability and scalability of the models. Catastrophic forgetting has also been observed in parameter-efficient fine-tuning methods, including LoRA, prompting recent studies to propose various approaches to mitigate this issue (Wang et al., 2024; Yang et al., 2024b; Ren et al., 2024; Yang et al., 2024a; Dou et al., 2024).

## 3 PROPOSED METHOD

### 3.1 MOTIVATIONS

In LoRA, there are some key insights on the *singular components* of the network parameters.

**Relationship between singular components and adaptation.** It is known that the dominant singular components with large singular values handle global information, while fine-grained singular components with smaller singular values and capture fine-grained details for full-rank matrix, such as weight matrix in deep learning. This distinction plays a crucial role in how the network processes tasks. To analyze how the pre-trained parameters are aligned with the fine-tuning task across various singular components, we decompose the pre-trained network parameters into singular component groups and measure the Fisher overlap (Kirkpatrick et al., 2017; Yao & Hansen, 2022; Qian et al., 2024) on each group for both tasks. A higher Fisher overlap indicates that the pre-trained network parameters are already aligned with the fine-tuning task and can be efficiently adapted by reusing them, as both tasks share knowledge and rely on a similar set of weights. Fig. 1 (a) illustrates the changes in the Fisher overlap for the pre-training and fine-tuning tasks, segmented from low to fine-grained singular components of the pre-trained network parameters. Notably, the pre-trained parameters reconstructed in the dominant singular value range exhibit a relatively high overlap ratio, while the overlap ratio decreases as the singular increases. This observation suggests that the dominant singular components of the pre-trained network parameters are already aligned to the fine-tuning task and can be reused for the fine-tuning task to a great extent; the higher-singular components become more task-specific and thus require a significant adaptation.

**Relationship between singular components and catastrophic forgetting.** The optimization of deep learning models, including LoRA, is to perform a Maximum A Posteriori (MAP) estimation of the network parameters $\theta$ on the training data. In transfer learning, the models are pre-trained using the pre-training dataset $D_A$ and fine-tuned on the fine-tuning dataset $D_B$. As revealed in Kirkpatrick et al. (2017), the posterior that needs to be maximized in MAP estimation is as follows:

$$p(\theta|\mathcal{D}_A, \mathcal{D}_B) = \frac{p(\mathcal{D}_B|\theta)p(\theta|\mathcal{D}_A)}{p(\mathcal{D}_B|\mathcal{D}_A)}, \tag{4}$$

where $\mathcal{D}_B$ is assumed to be independent of $\mathcal{D}_A$. By taking the logarithm of the posterior, the objective of MAP becomes as follows:

$$\theta^* = \underset{\theta}{\operatorname{argmax}} \log p(\theta|\mathcal{D}_A, \mathcal{D}_B) = \underset{\theta}{\operatorname{argmax}} \big[ \log p(\mathcal{D}_B|\theta) + \log p(\theta|\mathcal{D}_A) \big]. \tag{5}$$

The first term is the likelihood of $\mathcal{D}_B$ given the parameters and expressed as the loss function for the fine-tuning task. The second term represents the prior of the parameters given $\mathcal{D}_A$. During fine-tuning, we incorporate the posterior of the pre-trained task $p(\theta|\mathcal{D}_A)$ as the prior of the fine-tuning task. However, since the true posterior probability is intractable, it can be expressed as a function $f(\theta)$ and approximated using the Laplace approximation, a well-established method in Bayesian deep learning for handling intractable posteriors (Kirkpatrick et al., 2017; Ritter et al., 2018; Wang et al., 2021; Matena & Raffel, 2022; Gawlikowski et al., 2023). The Laplace approximation of the posterior is derived from a second-order Taylor expansion around its mode $\theta_0$ as:

$$\log p(\theta|\mathcal{D}_A) \simeq \log \hat{p}(\theta|\mathcal{D}_A) = f(\theta_0) - \frac{1}{2}(\theta - \theta_0)^\top F(\theta - \theta_0), \tag{6}$$

where $F$ is the Fisher information matrix (Fisher, 1922). During fine-tuning, if the prior probability of the parameters decreases while learning a new task, it implies that the pre-trained knowledge is not sufficiently preserved. The following theorem shows that the prior probability is upper-bounded by the singular values of the difference between pre-trained and fine-tuned parameters.

**Theorem 3.1.** *Let $\theta_0$ and $\theta$ be the pre-trained and fine-tuned weights, respectively. Then the log probability of the prior $\log p(\theta|\mathcal{D}_A)$ for fine-tuning task can be approximated using Laplace Approximation as $\log p(\theta|\mathcal{D}_A) \simeq \log \hat{p}(\theta|\mathcal{D}_A) = f(\theta_0) - \frac{1}{2}(\theta - \theta_0)^\top F(\theta - \theta_0)$. From this, the approximated log probability of the prior is upper-bounded as follows:*

$$\log \hat{p}(\theta|\mathcal{D}_A) \leq f(\theta_0) - \lambda_{\min}(F)\sqrt{\sum_{n=1}^{r} \sigma_n^2}, \tag{7}$$

*where $\lambda_{\min}(\cdot)$ indicates the smallest eigenvalue and $\sigma_n$ is $n$-th singular value of $\theta - \theta_0$.*

The proof is described in Appendix F. It is worth to note that the negligibility of higher-order terms in the Laplace approximation is a well-established literature in (Kass et al., 1990), and we provide details in Appendix G. According to Theorem 3.1, $\log \hat{p}(\theta|\mathcal{D}_A)$ is upper bounded by the singular values of the parameter difference, which is adapter in LoRA. Specifically, larger singular values of the adapter lead to a decrease in the posterior from the pre-training task, resulting in the loss of pre-trained knowledge, the phenomenon called *catastrophic forgetting*. The catastrophic forgetting problem undermines the strengths of pre-trained models and hinders their adaptability to new tasks, making it crucial to address in order to maintain scalability and reliability in transfer learning (Wang et al., 2024; Yang et al., 2024b; Ren et al., 2024; Yang et al., 2024a; Dou et al., 2024).

In stochastic optimization, however, the spectral norm of weight matrices grows rapidly (Zhai et al., 2023). It is commonly assumed that stochastic gradients at a certain point can be expressed as $g = \mu + \epsilon \in \mathbb{R}^{d \times d}$, where $\mu$ is the mean and $\epsilon$ is a random variable representing noise. The following proposition establishes a lower bound on the spectral norm of the ideal update $\|\Delta\|$.

**Proposition 3.2.** *From Zhai et al. (2023), it holds that:*

$$\|\Delta\| \geq \sqrt{d}\sqrt{1 - \frac{1}{d^2}\sum_{i,j=1}^{d} \frac{\omega_{i,j}^2}{\mu^2 i, j + \omega_{i,j}^2}}. \tag{8}$$

The noise second moment $\omega^2$ is typically in the order of $\mu^2$. Hence, Proposition 3.2 indicates that the spectral norm of the ideal update should be large, growing linearly with $\sqrt{d}$. Moreover, for large batch sizes we would have $\omega^2 \ll 1$, resulting in $\|\Delta\| \sim \sqrt{d}$. The proposition demonstrates that the spectral norm of weight matrices grows rapidly for large dimensions when equipped with adaptive optimizers. Therefore, in a general probabilistic optimization, the spectral norm is learned in the direction of increasing in the transfer learning including LoRA. This means that the largest singular value increases, which reduces the probability of the pre-trained knowledge in MAP and causes catastrophic forgetting. Fig. 1 (b) shows that the singular values of the adapter in LoRA increase significantly in the early stage of fine-tuning. This increase results in the performance degradation of the pre-trained task, which suggests that LoRA is also vulnerable to catastrophic forgetting.

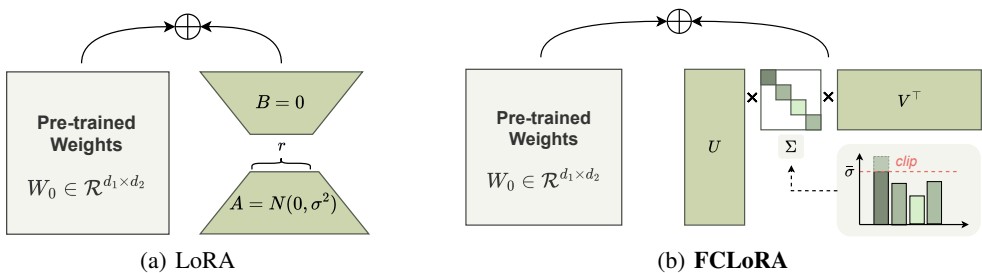

(a) LoRA  (b) **FCLoRA**

Figure 2: The architectures of LoRA and **FCLoRA**. **FCLoRA** employs parameterized SVD, where the learnable singular values are constrained by the pre-defined upper bound $\bar{\sigma}$.

### 3.2 LOW-RANK ADAPTATION WITH FINE-GRAINED COMPONENT INJECTION

Building upon these insights, we propose a **Lo**w **R**ank **A**daptation method with **F**ine-grained **C**omponent injection method, called **FCLoRA**, to adapt effectively for new tasks while retaining the pre-trained knowledge. Fig. 2 illustrates the architectures of the traditional LoRA and our proposed **FCLoRA**. While LoRA interprets $\Delta W$ as an adapter residual to the original pre-trained weights $W_0$, **FCLoRA** treats $\Delta W$ as an injected fine-grained singular component to $W_0$. To define $\Delta W$ as a matrix of learnable components with appropriate range of singular values, we parameterize the introduced modules in the form of singular value decomposition, based on the following common framework of the parameterized SVD-based LoRA (Zhang et al., 2023; Cao, 2024; Zhang et al., 2024), with our enhancement in managing the learnable singular components:

$$W = W_0 + \Delta W = W_0 + U\Sigma V^\top, \tag{9}$$

where $U \in \mathbb{R}^{d_1 \times r}$, $V^\top \in \mathbb{R}^{d_2 \times r}$ are parameterized left and right singular vectors, respectively, and $\Sigma \in \mathbb{R}^r$ contains the parameterized singular values $\{\sigma_n\}_{1 \leq n \leq r}$. $U, V$ are initialized with random $r$ singular vectors of $W_0$, or $U$ is initialized with zero and $V$ with a random Gaussian. Note that SVD on $W_0$ is performed only once before fine-tuning as in (Wang et al., 2024; Meng et al., 2024), and the actual operation does not involve any explicit decomposition or reconstruction of $W_0$ during the fine-tuning process. As mentioned earlier, we maintain the singular components of the introduced modules at an appropriate range. We set the lower bound of the singular values as zero according to the definition of SVD, ensuring the singular values to be non-negative. To enable the adapter to effectively inject fine-grained information for learning the new task, we constrain the injected singular values to lie below a pre-defined upper bound $\bar{\sigma}$, as expressed by the following equation:

$$\sigma_n = \min(\max(\sigma_n, 0), \bar{\sigma}), \tag{10}$$

where $\bar{\sigma}$ can hold the $q$-th quartile of the singular values of $W_0$, denoted as $\sigma^{(q)}$. To enforce the orthogonality of the singular vectors, i.e., $U^\top U = VV^\top = I$, we apply the regularization term as:

$$R(U, V) = \|U^\top U - I\| + \|VV^\top - I\| \tag{11}$$

where $I \in \mathbb{R}^{r \times r}$ indicates an identity matrix. This regularization term is controlled by the orthogonal regularization coefficient $\gamma$. We verify the orthogonality of the parameterized singular vectors in Appendix O.4. We present the training process in Algorithm 1 of Appendix J.

## 4 EXPERIMENTS

In this section, we empirically verify that **FCLoRA** efficiently adapts to the new task and improves the performance over other LoRA-based methods.

### 4.1 EXPERIMENTS ON NATURAL LANGUAGE UNDERSTANDING

**Experimental setup.** We evaluate **FCLoRA** on the General Language Understanding Evaluation (GLUE) benchmark (Wang et al., 2018a). Following Hu et al. (2021) and Zhang et al. (2023), we adopt the pre-trained RoBERTa$_{\text{base}}$ and DeBERTa$_{\text{base}}$ as the backbone models, respectively. We report Matthews correlation for CoLA, Spearman correlations for STS-B, and accuracy scores for the other tasks. The detailed descriptions are provided in Appendix M.1.2.

Table 1: Comparison of various methods with RoBERTa$_{base}$ on GLUE tasks with different random seeds. Full results with standard deviations are provided in Appendix M.1.4.

| Method | # Params | MNLI | SST-2 | CoLA | QQP | QNLI | RTE | MRPC | STS-B | Avg. |
|---|---|---|---|---|---|---|---|---|---|---|
| LoRA | 1.33M | 87.93 | 94.80 | 64.49 | 90.94 | 92.73 | 80.39 | 89.05 | 90.87 | 86.40 |
| AdaLoRA | 1.27M | 87.21 | 95.07 | 61.37 | 89.75 | 92.54 | 81.11 | 89.05 | 90.62 | 85.84 |
| PiSSA | 1.33M | **87.95** | 94.53 | 64.66 | 90.97 | 92.53 | 79.18 | 89.79 | 90.96 | 86.32 |
| rsLoRA | 1.33M | 85.26 | 92.35 | 65.17 | 70.76 | 92.48 | 79.54 | 89.05 | 90.88 | 83.19 |
| LoRA+ | 1.33M | 86.96 | 93.92 | 63.32 | 90.69 | 92.77 | 81.59 | 88.97 | 90.84 | 86.13 |
| MiLoRA | 1.33M | 87.88 | 94.69 | 64.31 | **91.02** | 92.96 | 81.35 | 89.30 | 90.96 | 86.56 |
| DoRA | 1.41M | 87.81 | 95.11 | 64.23 | 90.65 | 92.93 | 81.35 | 89.54 | 91.01 | 86.58 |
| **FCLoRA** | 1.33M | **87.95** | **95.37** | **64.79** | 90.76 | **93.09** | **83.15** | **90.32** | **91.22** | **87.08** |

Table 2: Comparison of various methods with DeBERTaV3$_{base}$ on GLUE tasks with different random seeds. The results for the baselines are copied from Zhang et al. (2023). Full results with standard deviations are provided in Appendix M.1.4.

| Method | # Params | MNLI | SST-2 | CoLA | QQP | QNLI | RTE | MRPC | STS-B | Avg. |
|---|---|---|---|---|---|---|---|---|---|---|
| Full FT | 184M | 89.90 | 95.63 | 69.19 | 92.40 | 94.03 | 83.75 | 89.46 | 91.60 | 88.09 |
| BitFit | 0.10M | 89.37 | 94.84 | 66.96 | 88.41 | 92.24 | 78.70 | 87.75 | 91.35 | 86.02 |
| HAdapter | 1.22M | 90.13 | 95.53 | 68.64 | 91.91 | 94.11 | 84.48 | 89.95 | 91.48 | 88.12 |
| PAdapter | 1.18M | 90.33 | 95.61 | 68.77 | 92.04 | 94.29 | 85.20 | 89.46 | 91.54 | 88.24 |
| LoRA$_{r=8}$ | 1.33M | 90.65 | 94.95 | 69.82 | 91.99 | 93.87 | 85.20 | 89.95 | 91.60 | 88.34 |
| AdaLoRA | 1.27M | **90.76** | 96.10 | 71.45 | 92.23 | 94.55 | 88.09 | 90.69 | 91.84 | 89.31 |
| **FCLoRA** | 1.33M | 90.36 | **96.33** | **71.49** | **92.33** | **94.57** | **89.41** | **91.58** | **92.19** | **89.78** |
| HAdapter | 0.61M | 90.12 | 95.30 | 67.87 | 91.65 | 93.76 | 85.56 | 89.22 | 91.30 | 87.93 |
| PAdapter | 0.60M | 90.15 | 95.53 | 69.48 | 91.62 | 93.98 | 84.12 | 89.22 | 91.52 | 88.04 |
| HAdapter | 0.31M | 90.10 | 95.41 | 67.65 | 91.54 | 93.52 | 83.39 | 89.25 | 91.31 | 87.60 |
| PAdapter | 0.30M | 89.89 | 94.72 | 69.06 | 91.40 | 93.87 | 84.48 | 89.71 | 91.38 | 87.90 |
| LoRA$_{r=2}$ | 0.33M | 90.30 | 94.95 | 68.71 | 91.61 | 94.03 | 85.56 | 89.71 | 91.68 | 88.15 |
| AdaLoRA | 0.32M | **90.66** | 95.80 | 70.04 | 91.78 | 94.49 | 87.36 | 90.44 | 91.63 | 88.86 |
| **FCLoRA** | 0.33M | **90.66** | **96.18** | **71.83** | **91.82** | 94.50 | **89.89** | **91.83** | **92.00** | **89.84** |

**Experimental results.** In Table 1, MiLoRA, which is closely related to our model, fails to achieve optimal performance due to information loss caused by directly modifying the fine-grained singular components. In contrast, motivated by the tendency of network parameter overlap, **FCLoRA** efficiently adapts to fine-tuning tasks by injecting the fine-grained singular components.

## 4.2 EXPERIMENTS ON QUESTION ANSWERING

**Experimental setup.** We evaluate **FCLoRA** on two question answering (QA) tasks: SQuADv1.1 (Rajpurkar, 2016) and SQuADv2.0 (Rajpurkar et al., 2018). Following Zhang et al. (2023), we fine-tune a DeBERTaV3$_{base}$ (He et al., 2021). We measured the performance using the Exact Match (EM) and F1 metrics. The detailed descriptions are provided in Appendix M.2.2.

**Experimental results.** Table 3 reports the experimental results of fine-tuning DeBERTa$_{base}$ on the QA task under four different budget settings: 0.08%, 0.16%, 0.32%, and 0.65% of the total pre-trained parameters. The proposed method outperformed the baselines across most settings, highlighting that fine-grained singular information can be effectively and efficiently adapted to new tasks

## 4.3 EXPERIMENTS ON COMMONSENSE REASONING

**Experimental setup.** We evaluate **FCLoRA** on the commonsense reasoning tasks. Following Hu et al. (2023), we amalgamate the training datasets from all 8 tasks to create the final training dataset and evaluate with individual testing for each task. We fine-tune LLaMA-7B (Touvron et al., 2023a) and LLaMA2-7B (Touvron et al., 2023b). The detailed descriptions are provided in Appendix M.3.2.

**Experimental results.** Table 4 reports the results on commonsense reasoning tasks. **FCLoRA** outperforms other methods, highlighting the effectiveness of fine-grained singular components of adapters even in larger models. The result suggests that fine-tuning with fine-grained singular components plays a crucial role in enhancing reasoning performance across diverse tasks.

Table 3: Comparison of various methods with DeBERTaV3$_{base}$ on SQuAD datasets

| Method | SQuADv1.1 | | | | SQuADv2.0 | | | |
|---|---|---|---|---|---|---|---|---|
| | 0.08% | 0.16% | 0.32% | 0.65% | 0.08% | 0.16% | 0.32% | 0.65% |
| Full FT* | 86.0 / 92.7 | | | | 85.4 / 88.4 | | | |
| HAdapter | 84.4/91.5 | 85.3/92.1 | 86.1/92.7 | 86.7/92.9 | 83.4/86.6 | 84.3/87.3 | 84.9/87.9 | 85.4/88.3 |
| PAdapter | 84.4/91.7 | 85.9/92.5 | 86.2/92.8 | 86.6/93.0 | 84.2/87.2 | 84.5/87.6 | 84.9/87.8 | 84.5/87.5 |
| LoRA | 86.4/92.8 | 86.6/92.9 | 86.7/93.1 | 86.7/93.1 | 84.7/87.5 | 83.6/86.7 | 84.5/87.4 | 85.0/88.0 |
| AdaLoRA | 87.2/93.4 | 87.5/93.6 | 87.5/93.7 | 87.6/93.7 | **85.6/88.7** | 85.7/88.8 | 85.5/88.6 | 86.0/88.9 |
| **FCLoRA** | **87.6/93.6** | **88.1/93.9** | **88.2/94.1** | **88.6/94.3** | 85.3/88.3 | **85.9/88.7** | **86.0/88.8** | **86.2/89.0** |

Table 4: Comparison of various methods with LLaMA on commonsense reasoning tasks

| Model | Method | #Params (%) | BoolQ | PIQA | SIQA | HellaSwag | WinoGrande | ARC-e | ARC-c | OBQA | Avg. |
|---|---|---|---|---|---|---|---|---|---|---|---|
| ChatGPT | - | - | 73.1 | 85.4 | 68.5 | 78.5 | 66.1 | 89.8 | 79.9 | 74.8 | 77.0 |
| LLaMA-7B | Prefix | 0.11 | 64.3 | 76.8 | 73.9 | 42.1 | 72.1 | 72.9 | 54.0 | 60.6 | 64.6 |
| | Series | 0.99 | 63.0 | 79.2 | 76.3 | 67.9 | 75.7 | 74.5 | 57.1 | 72.4 | 70.8 |
| | Parallel | 3.54 | 67.9 | 76.4 | 78.8 | 69.8 | 78.9 | 73.7 | 57.3 | 75.2 | 72.2 |
| | LoRA | 0.83 | 68.9 | 80.7 | 77.4 | 78.1 | 78.8 | 77.8 | 61.3 | 74.8 | 74.7 |
| | DoRA$^{\dagger}$ | 0.43 | 70.0 | 82.6 | 79.7 | 83.2 | 80.6 | 80.6 | 65.4 | 77.6 | 77.5 |
| | DoRA | 0.84 | 69.7 | 83.4 | 78.6 | 87.2 | 81.0 | 81.9 | 66.2 | 79.2 | 78.4 |
| | **FCLoRA** | 0.83 | 70.5 | 82.2 | 79.3 | 86.2 | 81.5 | 81.7 | 66.9 | 80.2 | **78.6** |
| LLaMA2-7B | LoRA | 0.83 | 69.8 | 79.9 | 79.5 | 83.6 | 82.6 | 79.8 | 64.7 | 81.0 | 77.6 |
| | DoRA$^{\dagger}$ | 0.43 | 72.0 | 83.1 | 79.9 | 89.1 | 83.0 | 84.5 | 71.0 | 81.2 | 80.5 |
| | DoRA | 0.84 | 71.8 | 83.7 | 76.0 | 89.1 | 82.6 | 83.7 | 68.2 | 82.4 | 79.7 |
| | **FCLoRA** | 0.83 | 73.2 | 82.9 | 79.8 | 91.9 | 83.0 | 85.2 | 71.6 | 82.6 | **81.3** |

## 5 DISCUSSIONS ON CATASTROPHIC FORGETTING

This section discusses how **FCLoRA** mitigates *catastrophic forgetting* compared to LoRA variants.

### 5.1 COMPARISON WITH OTHER LORA-BASED METHODS

The traditional LoRA freezes the whole pre-trained network parameters and learns adapters which are initialized to zero. PiSSA (Meng et al., 2024) and MiLoRA (Wang et al., 2024) adapt to new tasks by directly adjusting the $r$ highest/lowest singular values, respectively, making them highly relevant to **FCLoRA** in terms of employing singular values. However, existing methods suffer from catastrophic forgetting since: i) PiSSA and MiLoRA directly modify a subset of pre-trained parameters, leading to the loss of pre-trained knowledge, and ii) the lack of constraints on the singular values in the adapters during fine-tuning causes singular values to grow and results in reducing the posterior of the pre-trained task (see Theorem 3.1). In contrast, **FCLoRA** restricts the upper bound of the singular values of adapters during fine-tuning. This prevents the growth of spectral norms in adapters without incurring overhead, thereby effectively mitigating catastrophic forgetting.

### 5.2 SPECTRAL ANALYSIS OF $\Delta W$

Proposition 3.2 demonstrates that the spectral norm of large weight matrices increases rapidly when adaptive optimizers are applied. However, **FCLoRA** prevents this increase by restricting the range of the singular value of $\Delta W$ during fine-tuning, ensuring that the spectral norm does not grow excessively. To empirically verify this difference, Fig. 3 (a) illustrates the evolution of the spectral norm of $\Delta W$ across various methods during the fine-tuning. While LoRA and its variants tend to increase the spectral norm during fine-tuning, **FCLoRA** maintains a smaller spectral norm. This suggests that, unlike other LoRA-based methods, **FCLoRA** adapts to fine-tuning tasks by effectively incorporating new information with the fine-grained singular components.

### 5.3 EXPERIMENTS ON MITIGATING CATASTROPHIC FORGETTING

We previously demonstrated that the spectral norm of existing LoRA-based methods growth rapidly during fine-tuning. Fig. 3 (b) and (c) empirically show the accuracy and evaluation loss on the pre-trained task with the BookCorpus dataset. As the fine-tuning progresses, LoRA and its variants rapidly degrade the accuracy on the pre-trained task, dropping from the original performance of 0.6 to below 0.1, and the evaluation loss increases by more than 4 times. As mentioned earlier, without restrictions on the range of singular values of $\Delta W$ during fine-tuning, the model undergoes a rapid

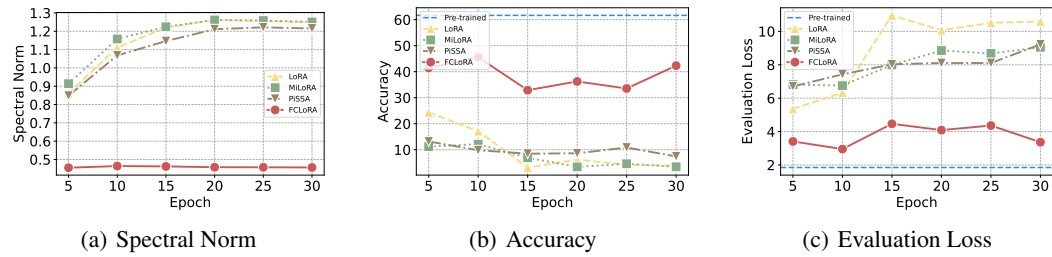

Figure 3: Changes during fine-tuning RoBERTa$_{\text{base}}$ on the MRPC dataset: (a) Spectral norm of $\Delta W$ in the query layer; (b, c) accuracy and evaluation loss on the pre-trained task (BookCorpus dataset).

growth of the spectral norm, leading to catastrophic forgetting of the pre-trained knowledge. In contrast, **FCLoRA** effectively mitigates catastrophic forgetting by restricting the range of singular values of $\Delta W$. This prevents excessive growth of the spectral norm, ensuring it remains at an appropriate level during fine-tuning and minimizing performance degradation on pre-trained tasks. We further validate the effectiveness of **FCLoRA** on mitigating catastrophic forgetting in Appendix N.

## 6 ADDITIONAL STUDIES

In Appendix O, we examine the effects of $\gamma$, $\bar{\sigma}$ and $r$ as the sensitivity analysis.

### 6.1 ABLATION STUDY ON THE INJECTED COMPONENTS

To analyze the influence of the injected components in **FCLoRA** on the performance of both pre-trained and fine-tuned knowledge, we conduct an ablation study on the following variants: i) 'LoRA' refers to the traditional LoRA; ii) 'LoRA$_{UV^\top}$' applies orthogonal regularization to the singular vectors without the singular values; iii) 'LoRA$_{SVD}$' initializes the singular values as ones, allowing them to be learnable from LoRA$_{UV^\top}$; and iv) '**FCLoRA**' refers to the proposed method. We measure the accuracy on both the fine-tuning tasks with MRPC and SST-2 from the GLUE task, and the pre-trained task with BookCorpus dataset. In Table 5, LoRA significantly sacrifices pre-training per-

Table 5: Ablation on the injected components

| Model | MRPC | | SST-2 | |
|---|---|---|---|---|
| | Acc.$_{\text{fine-tune}}$ | Acc.$_{\text{pre-train}}$ | Acc.$_{\text{fine-tune}}$ | Acc.$_{\text{pre-train}}$ |
| Pre-trained | - | 61.64 | - | 61.64 |
| LoRA | 89.05 | 3.77 | 94.81 | 32.35 |
| LoRA$_{UV^\top}$ | 89.22 | 3.12 | 94.75 | 39.39 |
| LoRA$_{SVD}$ | 89.62 | 17.57 | 95.03 | 49.65 |
| **FCLoRA** | **90.32** | **32.00** | **95.37** | **51.29** |

formance to adapt on fine-tuning task. For the MRPC dataset, accuracy on the pre-trained task drops from 61.64 to 3.77, while it achieves comparable accuracy on the fine-tuning task. LoRA$_{UV^\top}$ has limited expressiveness since its singular values are fixed, sacrificing either performance of pre-trained or fine-tuning task. LoRA$_{SVD}$ performs better due to its learnable singular values than LoRA$_{UV^\top}$. Notably, **FCLoRA** constrains the singular values to learn with the fine-grained singular components, ensuring both effective adaptation on fine-tuning task and retention of pre-trained information. For both datasets, **FCLoRA** achieves the best performance on both tasks.

## 7 CONCLUSION

We propose a novel low-rank adaptation method called **FCLoRA**, motivated by the following two rigorous analyses regarding the singular components of network parameters: i) We, for the first time, analyze LoRA via the Fisher overlap across the singular components. Specifically, the dominant singular components of pre-trained weights can be reused for fine-tuning tasks, whereas the fine-grained singular components are more task-specific and require significant adaptation. ii) The growth of singular values in the adapters directly causes catastrophic forgetting from the perspective of MAP estimation. From these analyses, we design **FCLoRA**, which injects the pre-trained model with fine-grained singular components. Experimental results show that **FCLoRA** achieves strong performance on fine-tuning tasks and successfully retains the pre-trained knowledge.

**Limitation.** Despite the advantages of **FCLoRA**, the optimal range of singular values to compose fine-grained singular components may vary across datasets, requiring further tuning in some cases.

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

## A    Reproducibility Statement

In an effort to ensure reproducibility, we report the description of dataset in Appendix M.1.1, Appendix M.2.1 and Appendix M.3.1. Also we report the best hyperparameters of our experiments in Appendix M.1.3, Appendix M.2.3 and Appendix M.3.3. Our **FCLoRA** code to reproduce the experiment can be found at `https://bit.ly/3ElHoYb`.

## B    Ethical Statement

We utilized publicly available datasets, including GLUE, SQuAD and commonsense reasoning, which are commonly employed in academic research, and all sources have been appropriately cited. This research does not involve any personal or confidential information, thereby eliminating concerns related to privacy. Our proposed approach and the resulting insights contribute to the advancement of artificial intelligence while adhering to principles of ethical innovation and responsibility.

## C    Broader Impact Statement

This paper presents work whose goal is to advance the field of Machine Learning. There are many potential societal consequences of our work, none which we feel must be specifically highlighted.

## D    Use of LLMs

In accordance with ICLR 2026 policy, we acknowledge the use of LLMs in the preparation of this paper. Their use was limited solely to improving translation accuracy and ensuring grammatical correctness.

## E    Formulation & Additional Visualization of Fisher Overlap

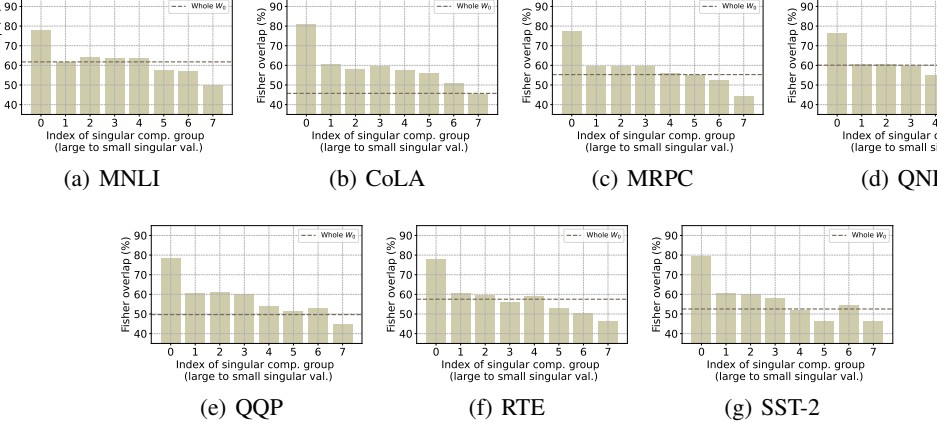

Figure 4: The Fisher overlap (Kirkpatrick et al., 2017) between the pre-trained task (BookCorpus) and the fine-tuning task (MRPC), evaluated over partial reconstructions of the pre-trained network parameters obtained by grouping singular components sorted from large to small according to their singular values.

Following Kirkpatrick et al. (2017), to examine whether different tasks solved by the same network rely on overlapping parameter subsets (see Fig. 1 (a)), we assessed the similarity of each task's Fisher information matrix. Specifically, we first computed the Fisher matrices for the two tasks, denoted by $F_1$ and $F_2$. We then normalized each matrix so that its trace was equal to 1, yielding $\hat{F}_1$ and $\hat{F}_2$. Next, we measure how closely these matrices aligned by computing the Fréchet distance, a

metric on positive-semidefinite matrices, given as:

$$d^2(\hat{F}_1, \hat{F}_2) = \frac{1}{2}\text{tr}\big(\hat{F}_1 + \hat{F}_2 - 2(\hat{F}_1\hat{F}_2)^{1/2}\big) = \frac{1}{2}\|\hat{F}_1^{1/2} - \hat{F}_2^{1/2}\|_F, \tag{12}$$

where this quantity lies between 0 and 1. We then define the *overlap* of the two tasks' Fisher matrices as $1 - d^2$. Hence, an overlap of 0 implies that the two tasks employ entirely distinct sets of parameters, whereas an overlap of 1 indicates that one Fisher matrix is simply a scaled version of the other (i.e., $F_1 = \alpha F_2$ for some $\alpha > 0$).

Then, to verify the Fisher overlap across the singular components of the pre-trained network parameters, we perform SVD on the pre-trained parameters of the RoBERTa$_{\text{base}}$ model, trained on multiple datasets including BookCorpus. We group the singular components sorted by their singular values and reconstruct partial versions of the parameters from each group. Fig. 4 shows the Fisher overlap computed for the BookCorpus task as well as for each task in the GLUE benchmark. Across all tasks, the Fisher overlap progressively decreases as we move from groups containing larger singular values to those with smaller ones, indicating that fine-tuning increasingly struggles to reuse the finer-grained partial reconstructions of the pre-trained parameters.

## F  PROOF OF THEOREM 3.1

**Theorem 3.1.** *Let $\theta_0$ and $\theta$ be the pre-trained and fine-tuned weights, respectively. Then the log probability of the prior $\log p(\theta|\mathcal{D}_A)$ for fine-tuning task can be approximated using Laplace Approximation as $\log p(\theta|\mathcal{D}_A) \simeq f(\theta_0) - \frac{1}{2}(\theta - \theta_0)^\top F(\theta - \theta_0)$. From this, the approximated log probability of the prior is upper-bounded as follows:*

$$\log \hat{p}(\theta|\mathcal{D}_A) \leq f(\theta_0) - \lambda_{\min}(F)\sqrt{\sum_{n=1}^{r} \sigma_n^2}, \tag{13}$$

*where $\lambda_{\min}(\cdot)$ indicates the smallest eigenvalue and $\sigma_n$ is $n$-th singular value of $\theta - \theta_0$.*

*Proof.* The optimization of neural networks can be considered as a process of estimating the network parameters $\theta$ through maximum a posteriori (MAP) estimation using the training data. This involves the pre-training dataset $\mathcal{D}_A$ and the fine-tuning dataset $\mathcal{D}_B$. The pre-trained weights are denoted as $\theta_0$, and the fine-tuned weights are represented as $\theta$.

The posterior to be maximized in the MAP estimation is formulated as:

$$p(\theta|\mathcal{D}_A, \mathcal{D}_B) = \frac{p(\mathcal{D}_B|\theta, \mathcal{D}_A)p(\theta|\mathcal{D}_A)}{p(\mathcal{D}_B|\mathcal{D}_A)} = \frac{p(\mathcal{D}_B|\theta)p(\theta|\mathcal{D}_A)}{p(\mathcal{D}_B|\mathcal{D}_A)}. \tag{14}$$

Taking a logarithm of the posterior, the MAP objective becomes:

$$\theta^* = \underset{\theta}{\text{argmax}} \log p(\theta|\mathcal{D}_A, \mathcal{D}_B) = \underset{\theta}{\text{argmax}}\big[\log p(\mathcal{D}_B|\theta) + \log p(\theta|\mathcal{D}_A)\big]. \tag{15}$$

Since the true posterior is intractable, we approximate the posterior using Laplace Approximation. $\log p(\theta|\mathcal{D}_A)$ can be expressed as a function $f(\theta)$ and approximated near the optimal point $f(\theta_0)$, where $\theta_0$ represents the pre-trained parameters, and $\nabla f(\theta_0) = 0$. Subsequently, a second-order Taylor expansion of $f(\theta)$ around $\theta_0$ is performed as follows:

$$\log p(\theta|\mathcal{D}_A) \simeq f(\theta_0) + \frac{1}{2}(\theta - \theta_0)\nabla^2 f(\theta_0)(\theta - \theta_0) = f(\theta_0) + \frac{1}{2}(\theta - \theta_0)^\top H(\theta - \theta_0), \tag{16}$$

where $H$ denotes the Hessian matrix of $f(\theta)$ evaluated at $\theta_0$. The expected value of the Hessian over the data distribution corresponds to the Fisher information matrix (FIM) $F$, defined as $F = -\mathbb{E}_{\mathcal{D}_A}[H]$. Following (MacKay, 1992; Kirkpatrick et al., 2017), we approximate the posterior as a Gaussian distribution with mean given by the parameters $\theta_0$ and a diagonal precision given by the diagonal of the Fisher information matrix $F$. Given this approximation, the log probability can be expressed as:

$$\log p(\theta|\mathcal{D}_A) \simeq \log \hat{p}(\theta|\mathcal{D}_A) = f(\theta_0) - \frac{1}{2}(\theta - \theta_0)^\top F(\theta - \theta_0), \tag{17}$$

where the $F$ is symmetric and positive semi-definite, i.e., for any vector $v$, $vFv^\top \geq 0$. Then using the singular value decomposition (SVD) on $\theta - \theta_0 = \Delta\theta = U\Sigma V^\top$ where $U \in \mathbb{R}^{d_1 \times r}$, $V \in \mathbb{R}^{r \times d_2}$, and $\Sigma \in \mathbb{R}^r$ with $\{\sigma_n\}_{1 \leq n \leq r}$. As $U, V$ are orthonormal singular vectors and $F$ is a positive semi-definite matrix,

$$\Delta\theta^\top F \Delta\theta \geq \lambda_{\min}(F)\|\Delta\theta\|_F = \lambda_{\min}(F)\|\Sigma\|_F. \tag{18}$$

Therefore, the log probability of the approximated prior for the fine-tuning task from the pre-trained task is upper bounded as:

$$\log \hat{p}(\theta|\mathcal{D}_A) = f(\theta_0) - \tfrac{1}{2}(\theta - \theta_0)^\intercal F(\theta - \theta_0) \leq f(\theta_0) - \lambda_{\min}(F)\,\|\Sigma\|_F$$

$$= f(\theta_0) - \lambda_{\min}(F)\sqrt{\sum_{n=1}^{r}\sigma_n^2} \tag{19}$$

$\square$

# G  WELL-ESTABLISHED PROPERTIES ON LAPLACE APPROXIMATION

## G.1  ERROR BOUND OF LAPLACE APPROXIMATION

In Bayesian inference, one of the most widely used methods for approximating the posterior distribution is the Laplace approximation (Kass et al., 1990; Kirkpatrick et al., 2017; Ritter et al., 2018; Wang et al., 2021; Matena & Raffel, 2022; Gawlikowski et al., 2023). This method expands the log-posterior function around its mode (i.e., the MAP estimate) using a Taylor series and retains terms up to the second order, thereby approximating the posterior distribution by a Gaussian distribution. In other words, higher-order terms beyond the quadratic expansion are discarded, and the resulting approximation error is generally limited and asymptotically negligible.

In particular, under standard regularity conditions, it is well established that the relative error of Laplace approximation is no worse than $\mathcal{O}_p(n^{-1})$ under standard regularity conditions, where $\mathcal{O}_p$ refers to stochastic boundedness (Kass et al., 1990; Bilodeau et al., 2023). This ensures that, as the sample size $n$ increases, the approximation error vanishes at the rate of $n^{-1}$. Moreover, in deep learning, where the number of training samples $n$ is typically very large, the accuracy of the Laplace approximation is further reinforced. Consequently, the Laplace approximation provides not only a practical tool but also a theoretically justified method for posterior approximation in both Bayesian inference and large-scale probabilistic modeling.

## G.2  DECAY RATE OF THE INTEGRAL OVER THE MODE-DISTANT REGION

Laplace approximation is applied when the target function is sharply concentrated around a single mode $\theta_0$ and decays rapidly as $\theta$ moves away from it. The method rewrites the integral in exponential form and then approximates the log-posterior by a second-order Taylor expansion around its mode. The region where $\theta - \theta_0$ is large corresponds to the tail of the function. The approximation error in this region should not be judged solely by the magnitude of $|\theta - \theta_0|$; rather, its actual contribution to the integral must be considered. Kass et al. (1990) rigorously demonstrates that this tail contribution is negligible. First, as the sample size increases, the likelihood function becomes increasingly peaked, so the posterior concentrates around the mode $\theta_0$. Second, the integral over regions distant from $\theta_0$ decays exponentially with $n$, that is, it is bounded by $\exp(-nc)$ for some $c > 0$. Consequently, the integral over the mode-distant region converges to zero, and the dominant contribution to the integral arises near the mode.

# H  EXPONENTIAL DECAY OF SINGULAR VALUES

To find the best possible $n$-dimensional subspace $V_n$ such that the closest approximation $v \in V_n$ to $W$ minimizes the error $\|W - v\|_X$, the definition of Kolmogorov $n$-width is formulated as follows:

$$d_n(W, X) = \inf_{\substack{V_n \subset X \\ \dim V_n = n}} \inf_{v \in V_n} \|W - v\|_X, \tag{20}$$

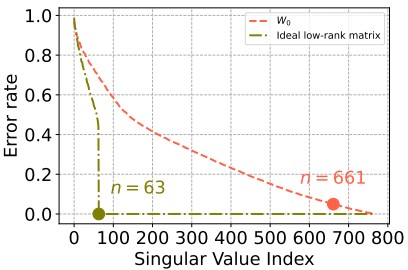

Figure 5: The error rate of the normalized singular values for: (i) the final output projection layer weights $W_0$ in the self-attention mechanism of DeBERTaV3$_{\text{base}}$, and (ii) an ideal low-rank matrix with rank $r = 64$. The marker indicates the $n$-value where the approximation error reaches 5%.

where $V_n$ is $n$-dimensional subspace of $X$, $v$ is an element from the subspace $V_n$. 'inf' stands for infimum. When using the Frobenius norm (or spectral norm) with matrices, the Kolmogorov $n$-width is computed by the singular values of $W$ as follows:

$$d_n(W, X) = \sigma_{n+1}, \tag{21}$$

where $\sigma_{n+1}$ is the $(n + 1)$-th largest singular value of the matrix $W$. The Kolmogorov $n$-width measures how well a set $W$ can be approximated by an $n$-dimensional subspace. In other words, it represents the minimal maximum error when approximating with an $n$-dimensional subspace. Then we can determine the optimal dimensionality needed to achieve a desired approximation accuracy.

If the singular values decrease rapidly, $W$ can be well approximated even for small $n$, and the Kolmogorov $n$-width also decreases quickly. Therefore, the singular value decay rate $\alpha$, which plays a pivotal role in determining how effectively a matrix can be approximated, is commonly modeled by an exponential decay function as follows:

$$\sigma'_n = Ce^{-\alpha n}, \tag{22}$$

where $\sigma'_n$ represents the $n$-th modeled singular values, $C > 0$ is a constant, and $\alpha > 0$ is the decay rate. When the decay rate $\alpha$ is low, the singular values decrease gradually, resulting in large errors when approximating with the same $n$ dimensions. To minimize the approximation errors, a larger $n$ is required, indicating that significant information is contained in the lower singular values.

**Empirical analysis of the Kolmogorov $n$-width.** To empirically analyze the Kolmogorov $n$-width of the pre-trained language model, we present error rates based on low-rank approximation under the same conditions as shown in Fig. 1 (b) of Introduction. The formulation of error rates $E_W(n)$ is as follows:

$$E_W(n) = \left( \frac{\|W - v\|_F}{\|W\|_F} \right) \times 100\%, \tag{23}$$

where $W$ is the original matrix and $v$ is the approximated matrix obtained by truncating the SVD to rank $n$. The error rates for the pre-trained model and the ideal low-rank matrix are presented in Fig. 5, with markers indicating the $n$-value where the error rate reaches 5%. For the ideal low-rank matrix, the rank at which the error rate reaches 5% is 63. This suggests that the matrix has a low-dimensional structure, with the most important information concentrated in the top singular values. The lower singular values have little effect on the approximation and can be considered noise. In contrast, for the pre-trained model, the $n$-value required to reach 95% approximation is 661, which is significantly larger than ideal row rank matrix. This indicates that the data is complex and high-dimensional, and the lower singular values contain important information rather than merely noise.

# I  DISCUSSION ON CONSTRAINING THE FROBENIUS NORM OF $\Delta W$

Theorem 3.1 provides an upper bound of approximated posterior based on the Frobenius norm of $\Delta\theta$ and the minimum eigenvalue of the Fisher Information Matrix (FIM), $\lambda_{\min}(F)$. However, simply constraining the Frobenius norm is not sufficient. This is because the FIM $F$ encodes information

about *important directions* about the pre-trained task in the parameter space. Specifically, Equation (6) provides the approximated log posterior of pre-trained task using Laplace approximation. The FIM $F$ is a positive semi-definite matrix. The quadratic term quantifies how much the parameter shift $\Delta\theta = \theta - \theta_0$ affects the output distribution of the original task. In low-rank update methods like LoRA, the parameter update can be expressed as $\Delta\theta = U\Sigma V^\top$. Due to the low-rank nature of the update, the singular values $\sigma_i$ often become concentrated in a small number of singular directions.

Even for the same Frobenius norm of $\Delta\theta$, if $\sigma_n$ is heavily concentrated in certain directions, and those directions align with sensitive ones in the FIM (i.e., those with large eigenvalues), the penalty term $\Delta\theta^\top F \Delta\theta$ can become significantly larger (not $\Delta\theta^\top \Delta\theta$). As a result, the approximated posterior decreases more sharply, which intensifies catastrophic forgetting on the pre-trained task. Therefore, the Frobenius norm merely controls the total magnitude of change, but does not prevent the change from being concentrated in directions that are crucial for the original task. In contrast, **FCLoRA** explicitly controls individual singular values via clipping, thereby directly limiting updates in directions where the model is most sensitive. This goes beyond a simple norm constraint and introduces a structural bias that suppresses updates in forgetting-prone directions. Therefore, while the Frobenius norm constrains only the total size of the update, **FCLoRA** enables fine-grained control over which components grow, effectively mitigating catastrophic forgetting in a more targeted and principled way. This is also well-described in Kirkpatrick et al. (2017), where L2-norm cannot mitigate catastrophic forgetting because important parameters from previous tasks are not adequately preserved. The standard L2-norm regularization treats all parameters uniformly, failing to account for their varying importance, and thus cannot effectively mitigate this problem.

## J  ALGORITHM OF **FCLoRA**

In this section, we summarize the detailed algorithm of **FCLoRA** in Algorithm 1.

---

**Algorithm 1** How to train **FCLoRA**

---

**Input:** Dataset $\mathcal{D}$; total iterations $T$; learning rate $\eta$; $\gamma, \bar{\sigma}$.
**for** $t = 1, \ldots, T$ **do**
  $\Sigma_k^{(t)} = \min(\max(\Sigma_k^{(t)}, 0), \bar{\sigma})$
  $W_k^{(t)} = W_0 + U_k^{(t)} \Sigma_k^{(t)} (V_k^{(t)})^\top$
  Update $U_k^{(t+1)} = U_k^{(t)} - \eta \nabla_{U_k}(\mathcal{L}(U_k^{(t)}, \Sigma_k^{(t)}, V_k^{(t)}) + \gamma R(U_k^{(t)}, V_k^{(t)}))$
  Update $V_k^{(t+1)} = V_k^{(t)} - \eta \nabla_{V_k}(\mathcal{L}(U_k^{(t)}, \Sigma_k^{(t)}, V_k^{(t)}) + \gamma R(U_k^{(t)}, V_k^{(t)}))$
  Update $\Sigma_k^{(t+1)} = \Sigma_k^{(t)} - \eta \nabla_{\Sigma_k} \mathcal{L}(U_k^{(t)}, \Sigma_k^{(t)}, V_k^{(t)})$
**end**
**Output:** The fine-tuned parameters $\{U^{(T)}, \Sigma^{(T)}, V^{(T)}\}$; $W^{(T)} = W_0 + U^{(T)} \Sigma^{(T)} (V^{(T)})^\top$.

---

## K  LoRA WITH DISCRETE FOURIER TRANSFORM

Table 6: Comparison of various methods with RoBERTa$_{\text{base}}$ on GLUE tasks with the experimental setup from Gao et al. (2024). The results for the baselines are copied from Gao et al. (2024). The best performance is set in **bold**, and the second best is set in underline.

| Method | # Params | SST-2 | MRPC | CoLA | QNLI | RTE | STS-B | Avg. |
|---|---|---|---|---|---|---|---|---|
| FT | 125M | $94.8_{\pm0.2}$ | $90.2_{\pm0.6}$ | $63.6_{\pm2.6}$ | $92.8_{\pm0.2}$ | $78.7_{\pm0.7}$ | $91.2_{\pm0.5}$ | 85.2 |
| BitFit | 0.1M | $93.7_{\pm0.3}$ | $\mathbf{92.7_{\pm0.6}}$ | $62.0_{\pm1.9}$ | $91.8_{\pm0.2}$ | $\underline{81.5}_{\pm0.8}$ | $90.8_{\pm0.6}$ | $\underline{85.4}$ |
| BitFit | 0.1M | $93.7_{\pm0.3}$ | $\mathbf{92.7_{\pm0.6}}$ | $62.0_{\pm1.9}$ | $91.8_{\pm0.2}$ | $\underline{81.5}_{\pm0.8}$ | $90.8_{\pm0.6}$ | $\underline{85.4}$ |
| Adpt$^D$ | 0.3M | $94.2_{\pm0.1}$ | $88.5_{\pm1.1}$ | $60.8_{\pm1.1}$ | $93.1_{\pm0.4}$ | $71.5_{\pm2.7}$ | $89.7_{\pm0.7}$ | 83.0 |
| Adpt$^D$ | 0.9M | $94.7_{\pm0.3}$ | $88.4_{\pm0.1}$ | $62.6_{\pm0.9}$ | $93.0_{\pm0.2}$ | $75.9_{\pm0.4}$ | $90.3_{\pm0.2}$ | 84.2 |
| LoRA | 0.3M | $\mathbf{95.1_{\pm0.2}}$ | $89.7_{\pm0.7}$ | $63.4_{\pm1.0}$ | $\underline{93.3}_{\pm0.2}$ | $78.4_{\pm0.2}$ | $\mathbf{91.5_{\pm0.2}}$ | 85.2 |
| AdaLoRA | 0.3M | $94.5_{\pm0.2}$ | $88.7_{\pm0.5}$ | $62.0_{\pm0.6}$ | $93.1_{\pm0.3}$ | $81.0_{\pm0.9}$ | $90.5_{\pm0.2}$ | 85.0 |
| DyLoRA | 0.3M | $94.3_{\pm0.2}$ | $89.5_{\pm0.6}$ | $61.1_{\pm0.9}$ | $92.2_{\pm0.4}$ | $78.7_{\pm1.0}$ | $91.1_{\pm0.3}$ | 84.5 |
| FourierFT | 0.024M | $94.2_{\pm0.3}$ | $90.0_{\pm0.3}$ | $\underline{63.8}_{\pm1.6}$ | $92.2_{\pm0.1}$ | $79.1_{\pm0.2}$ | $90.8_{\pm0.4}$ | 85.0 |
| **FCLoRA** | 0.3M | $\underline{95.0}_{\pm0.2}$ | $\underline{90.6}_{\pm1.1}$ | $\mathbf{65.1_{\pm0.1}}$ | $\mathbf{93.4_{\pm0.2}}$ | $\mathbf{83.0_{\pm1.0}}$ | $\underline{91.3}_{\pm0.1}$ | **86.4** |

Discrete Fourier Transform (DFT) (Briggs & Henson, 1995) converts a finite sequence of equally-spaced samples of a function into a same-length sequence of equally-spaced samples of the discrete-time Fourier transform (DTFT), which is a complex-valued function of frequency. Recent studies have proposed the DFT-based parameter efficient fine-tuning method. FouRA (Borse et al., 2024) transforms the hidden vectors in the latent space into the singular domain using DFT and compute LoRA, followed by reconstruction through inverse DFT. FourierFT (Gao et al., 2024) considers the adapter as a 2D spatial-domain matrix and transforms into a 2D DFT spectrum. Therefore, existing methods have focused on DFT-based approaches to either flexibly select adapter ranks depending on the input or reduce the number of parameters. On the other hand, **FCLoRA** leverages a parameterized SVD in the parameter space, and injects the fine-grained singular components with upper bounded singular values to achieve effective adaptation and mitigation of catastrophic forgetting.

We conduct additional experiments following experimental setup of FourierFT (Gao et al., 2024), one of the LoRA variants that employs the concept of DFT. Strictly following (Gao et al., 2024), we fine-tune RoBERTa$_{\text{base}}$, adapting only the query and value projection matrices using LoRA. We maintain the same experimental settings as the original work, while only searching for the learning rate from $\{1 \times 10^{-3}, 8 \times 10^{-4}, 6 \times 10^{-4}, \}$, $\gamma$ from $\{1 \times 10^{-2}, 3 \times 10^{-3}, 1 \times 10^{-3}\}$, and $\bar{\sigma}$ from $\{\sigma^{(1)}, \sigma^{(2)}, \sigma^{(3)}, \sigma^{(4)}\}$ and the results are reported in Table 6.

## L HOW DOES THE ADAPTERS $\Delta W$ COMPARED TO $W$?

Table 7: The Frobenius norm of $U^\top W V$, where $U$ and $V$ are the left and right top $r$ singular vector directions of either: (1) $\Delta W_q$, (2) $W_q$, or (3) a random matrix. (4) The Frobenius norm of $U^\top \Delta W V$, where $U$ and $V$ are from $W_q$. (5) The Frobenius norm of $\Delta W$. (6,7) The introduced factors. The weights are taken from the last query layer of RoBERTa$_{\text{base}}$, fine-tuned on STS-B dataset with $r = 8$.

| Model | $\|U^\top W V\|_F$ | | | $\|U_{W_q}^\top \Delta W V_{W_q}\|_F$ | $\|\Delta W\|_F$ | Factor$_{W \to \Delta W}$ | Factor$_{\Delta W \to W}$ |
|---|---|---|---|---|---|---|---|
| | $\Delta W_q$ | $W_q$ | Random | | | | |
| LoRA | 0.48 | 11.22 | 0.32 | 0.16 | 3.81 | 7.94 | 23.82 |
| PiSSA | 0.38 | 11.22 | 0.35 | 0.11 | 2.49 | 6.54 | 22.60 |
| MiLoRA | 0.45 | 11.22 | 0.35 | 0.08 | 3.11 | 6.91 | 38.86 |
| **FCLoRA** | 0.36 | 11.22 | 0.38 | 0.03 | 0.94 | 2.60 | 31.18 |

We explore the relationship between $\Delta W$ and $W$ by measuring the correlation between $\Delta W$ and $W$ as well as the magnitude of $\Delta W$ in comparison to its corresponding directions in $W$. To do so, we introduce two key factors:

- Factor$_{W \to \Delta W}$ is a factor formulated as $\|\Delta W\|_F / \|U_{\Delta W}^\top W V_{\Delta W}\|_F$, which indicates the ratio of the norm of difference over the norm of projected $W$ on the $r$-dimensional subspace of $\Delta W$. This factor is also called *amplification factor* (Hu et al., 2021), measuring how the new information of $\Delta W$ is related to the existing information of $W$. A larger ratio refers that the task-specific information of $W$ has been amplified in $\Delta W$.

- Factor$_{\Delta W \to W}$ is a factor formulated as $\|\Delta W\|_F / \|U_W^\top \Delta W V_W\|_F$, which is the ratio of the norm of difference over the norm of projected $\Delta W$ on the $r$-dimensional subspace of $W$. It indicates the extent to which the change aligns with $W$. A larger ratio refers that $\Delta W$ has learned new information that is not present in $W$.

Following Hu et al. (2021), we project $W$ onto the $r$-dimensional subspace of $\Delta W$ by computing $U^\top W V$, where $U$ and $V$ are the left and right singular vectors of $\Delta W$, $W$, and the random matrix. Additionally, we project $\Delta W$ onto the subspace of $W$ by computing $U^\top \Delta W V$. As shown in Table 7, **FCLoRA** and other methods exhibit similar Frobenius norms when $W$ is projected onto the subspace of $\Delta W$, $W$ and random matrix. However, compared to the baselines, the projection of $\Delta W$ onto the subspace of $W$ in **FCLoRA** shows the lowest correlation with a value of 0.02, which is less than half of the smallest baseline. This suggests that **FCLoRA** processes the existing information in $W$ similarly to other methods, while being better at learning independent new information without relying on the existing information in $W$. Furthermore, considering the Frobenius norm of

$\Delta W$, both LoRA and PiSSA exhibit a large $\text{Factor}_{W \to \Delta W}$ and a small $\text{Factor}_{\Delta W \to W}$, indicating that $\Delta W$ primarily amplifies information already present in $W$. MiLoRA also shows a large $\text{Factor}_{\Delta W \to W}$, but this results from the large magnitude of $\Delta W$, leading to significant changes from the pre-trained weights. In contrast, **FCLoRA** exhibits a relatively small $\text{Factor}_{W \to \Delta W}$ of 2.60 but a large $\text{Factor}_{\Delta W \to W}$ of 31.18. Then we can cal Given the small magnitude of $\Delta W$, this indicates that **FCLoRA** stands out for its ability to learn new information that is not already in $W$ with minimal deviation from the pre-trained weights.

## M EXPERIMENTAL SETUP

### M.1 NATURAL LANGUAGE UNDERSTANDING

#### M.1.1 DATASET DESCRIPTION

We describe the benchmark datasets of GLUE (Wang et al., 2018a) below [1].

- **CoLA.** The Corpus of Linguistic Acceptability (Warstadt et al., 2019) provides a dataset of English sentences, where each sentence is judged for grammatical acceptability based on data from books and journal articles. The objective is a binary classification to determine whether a sentence is grammatically correct or incorrect. The dataset consists of 8.5k samples for training, 1k samples for validation, and 1k samples for test.

- **SST-2.** The Stanford Sentiment Treebank (Socher et al., 2013) includes sentences from movie reviews, along with human-provided sentiment annotations. The goal is to classify the sentiment of each sentence as either positive or negative. The dataset consists of 67k samples for training, 872 samples for validation, and 1.8k samples for test.

- **MRPC.** The Microsoft Research Paraphrase Corpus (Dolan & Brockett, 2005) contains pairs of sentences automatically extracted from online news sources. Human annotators label each pair, and the task is to identify whether the two sentences in a pair convey the same meaning. The dataset consists of 3.7k samples for training, 408 samples for validation, and 1.7k samples for test.

- **QQP.** The Quora Question Pairs dataset (Chen et al., 2018) consists of question pairs taken from Quora, a community-driven question-and-answer platform. The task is to determine if two given questions are semantically identical. The dataset consists of 364k samples for training, 40k samples for validation, and 391k samples for test.

- **MNLI.** The Multi-Genre Natural Language Inference Corpus (Williams et al., 2017) includes sentence pairs with textual entailment annotations collected through crowdsourcing. Given a premise and a hypothesis, the task is to predict whether the premise entails the hypothesis, contradicts it, or is neutral. The dataset includes both in-domain and cross-domain evaluations using a hidden test set. The dataset consists of 393k samples for training, 20k samples for validation, and 20k samples for test.

- **QNLI.** The Question-Answering Natural Language Inference dataset (Wang et al., 2018b) consists of question-paragraph pairs from which an answer must be found. The task involves determining whether a specific sentence from the paragraph answers the corresponding question. The dataset consists of 108k samples for training, 5.7k samples for validation, and 5.7k samples for test.

- **RTE.** The Recognizing Textual Entailment dataset (Bentivogli et al., 2009) comes from a series of annual challenges focusing on textual entailment. The task is to classify sentence pairs as either entailment or non-entailment. The dataset consists of 2.5k samples for training, 276 samples for validation, and 3k samples for test.

- **STS-B.** The Semantic Textual Similarity Benchmark (Cer et al., 2017) features sentence pairs drawn from various sources, including news headlines and image captions, with human-assigned similarity scores. The task is a regression problem where the model must predict a similarity score ranging from 0 to 5. The dataset consists of 7k samples for training, 1.5k samples for validation, and 1.4k samples for test.

---

[1] https://huggingface.co/datasets/nyu-mll/glue

### M.1.2 EXPERIMENTAL SETUP

We evaluate **FCLoRA** on the General Language Understanding Evaluation (GLUE) benchmark (Wang et al., 2018a), which includes 3 categories of natural language understanding tasks: i) single-sentence (CoLA and SST-2); ii) similarity and paraphrasing (MRPC, QQP, and STS-B); iii) natural language inference tasks (MNLI, QNLI, and RTE). For a fair comparison, following Hu et al. (2021), we adopt the pre-trained RoBERTa$_{\text{base}}$ as the backbone model. We use 1 GPU of NVIDIA RTX A6000 for experiments. We report Matthews correlation for CoLA, Spearman correlations for STS-B, and accuracy scores for the other tasks. We conduct the experiments in Huggingface Framework [2].

### M.1.3 HYPERPARAMETERS

Table 8: Best hyperparameters for **FCLoRA** in natural language understanding for RoBERTa$_{\text{base}}$

| Dataset | Learning rate | Batch size | #Epochs | Metric | $\bar{\sigma}$ | $\gamma$ | How to initialize $U, V$ |
|---|---|---|---|---|---|---|---|
| CoLA | $4 \times 10^{-4}$ | 32 | 25 | Matthews correlation | $\sigma^{(2)}$ | $3 \times 10^{-2}$ | random $r$ singular vectors |
| MNLI | $5 \times 10^{-4}$ | 32 | 7 | Accuracy | $\sigma^{(2)}$ | $1 \times 10^{-1}$ | 0, random Gaussian |
| MRPC | $4 \times 10^{-4}$ | 16 | 30 | Accuracy | $\sigma^{(2)}$ | $1 \times 10^{-2}$ | random $r$ singular vectors |
| QNLI | $4 \times 10^{-4}$ | 32 | 5 | Accuracy | $\sigma^{(1)}$ | $1 \times 10^{-1}$ | 0, random Gaussian |
| QQP | $5 \times 10^{-4}$ | 32 | 5 | Accuracy | $\sigma^{(1)}$ | $1 \times 10^{-3}$ | 0, random Gaussian |
| RTE | $5 \times 10^{-4}$ | 32 | 50 | Accuracy | $\sigma^{(4)}$ | $5 \times 10^{-2}$ | 0, random Gaussian |
| SST-2 | $5 \times 10^{-4}$ | 32 | 24 | Accuracy | $\sigma^{(4)}$ | $1 \times 10^{-1}$ | 0, random Gaussian |
| STS-B | $4 \times 10^{-4}$ | 32 | 25 | Pearson correlation | $\sigma^{(1)}$ | $1 \times 10^{-1}$ | 0, random Gaussian |

Table 9: Best hyperparameters for **FCLoRA** in natural language understanding for DeBERTaV3$_{\text{base}}$ with $r = 8$

| Dataset | Learning rate | Batch size | #Epochs | Metric | $\bar{\sigma}$ | $\gamma$ |
|---|---|---|---|---|---|---|
| CoLA | $8 \times 10^{-4}$ | 32 | 25 | Matthews correlation | $\sigma^{(4)}$ | $5 \times 10^{-1}$ |
| MNLI | $5 \times 10^{-4}$ | 32 | 7 | Accuracy | $\sigma^{(3)}$ | $1 \times 10^{-2}$ |
| MRPC | $1 \times 10^{-3}$ | 32 | 30 | Accuracy | $\sigma^{(3)}$ | $5 \times 10^{-1}$ |
| QNLI | $5 \times 10^{-4}$ | 32 | 5 | Accuracy | $\sigma^{(3)}$ | $1 \times 10^{-1}$ |
| QQP | $8 \times 10^{-4}$ | 32 | 5 | Accuracy | $\sigma^{(1)}$ | $1 \times 10^{-2}$ |
| RTE | $1.2 \times 10^{-3}$ | 32 | 50 | Accuracy | $\sigma^{(3)}$ | $1 \times 10^{-1}$ |
| SST-2 | $8 \times 10^{-4}$ | 32 | 24 | Accuracy | $\sigma^{(4)}$ | $1 \times 10^{-1}$ |
| STS-B | $2.2 \times 10^{-3}$ | 32 | 25 | Pearson correlation | $\sigma^{(2)}$ | $5 \times 10^{-1}$ |

Table 10: Best hyperparameters for **FCLoRA** in natural language understanding for DeBERTaV3$_{\text{base}}$ with $r = 2$

| Dataset | Learning rate | Batch size | #Epochs | Metric | $\bar{\sigma}$ | $\gamma$ |
|---|---|---|---|---|---|---|
| CoLA | $8 \times 10^{-4}$ | 32 | 25 | Matthews correlation | $\sigma^{(4)}$ | $1.1 \times 10^{-1}$ |
| MNLI | $5 \times 10^{-4}$ | 32 | 7 | Accuracy | $\sigma^{(3)}$ | $1 \times 10^{-1}$ |
| MRPC | $1 \times 10^{-3}$ | 32 | 30 | Accuracy | $\sigma^{(2)}$ | $1 \times 10^{-2}$ |
| QNLI | $7 \times 10^{-4}$ | 32 | 5 | Accuracy | $\sigma^{(3)}$ | $1 \times 10^{-1}$ |
| QQP | $8 \times 10^{-4}$ | 32 | 5 | Accuracy | $\sigma^{(1)}$ | $5 \times 10^{-3}$ |
| RTE | $1.2 \times 10^{-3}$ | 32 | 50 | Accuracy | $\sigma^{(3)}$ | $1 \times 10^{-1}$ |
| SST-2 | $8 \times 10^{-4}$ | 32 | 24 | Accuracy | $\sigma^{(4)}$ | $5 \times 10^{-1}$ |
| STS-B | $2.2 \times 10^{-3}$ | 32 | 25 | Pearson correlation | $\sigma^{(4)}$ | $6 \times 10^{-1}$ |

To tune **FCLoRA** with RoBERTa$_{\text{base}}$, we search for the learning rate from $\{4 \times 10^{-4}, 5 \times 10^{-4}\}$, $\bar{\sigma}$ from $\{\sigma^{(2)}, \sigma^{(3)}, \sigma^{(4)}\}$ and $\gamma$ from $\{1 \times 10^{-1}, 7 \times 10^{-2}, 5 \times 10^{-2}, 3 \times 10^{-2}, 1 \times 10^{-2}, 1 \times 10^{-3}\}$.

---

[2]https://github.com/huggingface/transformers

The learnable singular vectors $U/V$ can be initialized as i) random $r$ singular vectors of $W_0$, ii) $U$ with zeros, $V$ with random Gaussian initialization. To tune **FCLoRA** with DeBERTaV3$_{\text{base}}$, we search $\bar{\sigma}$ from $\{\sigma^{(2)}, \sigma^{(3)}, \sigma^{(4)}\}$ and $\gamma$ from $\{1 \times 10^{-1}, 1.1 \times 10^{-1}, 1 \times 10^{-2}, 5 \times 10^{-1}, 6 \times 10^{-1}\}$. The learnable singular vectors $U/V$ are be initialized as $U$ with zeros and $V$ with random Gaussian initialization. We report the best hyperparameters of **FCLoRA** in Table 8 to Table 10.

### M.1.4 EXPERIMENTAL RESULT WITH STANDARD DEVIATIONS

We report the experimental results on GLUE tasks with standard deviation of Roberta$_{\text{base}}$ and DeBERTa$_{\text{base}}$V3 in Table 11 and Table 12, respectively. Note that the results for Table 12 are copied from Zhang et al. (2023), the results with standard deviation are reported only for **FCLoRA**.

Table 11: Comparison of various methods on GLUE tasks with different random seeds.

| Method | MNLI | SST-2 | CoLA | QQP | QNLI | RTE | MRPC | STS-B | Avg. |
|---|---|---|---|---|---|---|---|---|---|
| LoRA | $87.93_{\pm 0.15}$ | $94.80_{\pm 0.11}$ | $64.49_{\pm 0.64}$ | $90.94_{\pm 0.04}$ | $92.73_{\pm 0.11}$ | $80.39_{\pm 0.74}$ | $89.05_{\pm 0.12}$ | $90.87_{\pm 0.08}$ | 86.40 |
| PiSSA | $87.95_{\pm 0.01}$ | $94.53_{\pm 0.19}$ | $64.66_{\pm 0.98}$ | $90.97_{\pm 0.05}$ | $92.53_{\pm 0.14}$ | $79.18_{\pm 0.68}$ | $89.79_{\pm 1.41}$ | $90.96_{\pm 0.08}$ | 86.32 |
| AdaLoRA | $87.21_{\pm 0.05}$ | $95.07_{\pm 0.32}$ | $61.37_{\pm 1.01}$ | $89.75_{\pm 0.10}$ | $92.54_{\pm 0.08}$ | $81.11_{\pm 0.85}$ | $89.05_{\pm 0.31}$ | $90.62_{\pm 0.11}$ | 85.84 |
| rsLoRA | $85.26_{\pm 3.34}$ | $92.35_{\pm 0.33}$ | $65.17_{\pm 0.82}$ | $70.76_{\pm 10.71}$ | $92.48_{\pm 0.27}$ | $79.54_{\pm 1.33}$ | $89.05_{\pm 0.31}$ | $90.88_{\pm 0.13}$ | 83.19 |
| LoRA+ | $86.96_{\pm 0.65}$ | $93.92_{\pm 0.00}$ | $63.32_{\pm 0.69}$ | $90.69_{\pm 0.07}$ | $92.77_{\pm 0.01}$ | $81.59_{\pm 1.18}$ | $88.97_{\pm 0.35}$ | $90.84_{\pm 0.14}$ | 86.13 |
| DoRA | $87.81_{\pm 0.04}$ | $95.11_{\pm 0.19}$ | $64.23_{\pm 0.10}$ | $90.65_{\pm 0.11}$ | $92.93_{\pm 0.10}$ | $81.35_{\pm 0.95}$ | $89.54_{\pm 0.23}$ | $91.01_{\pm 0.16}$ | 86.58 |
| MiLoRA | $87.88_{\pm 0.11}$ | $94.69_{\pm 0.30}$ | $64.31_{\pm 0.97}$ | $91.02_{\pm 0.04}$ | $92.96_{\pm 0.21}$ | $81.35_{\pm 1.23}$ | $89.30_{\pm 0.12}$ | $90.96_{\pm 0.05}$ | 86.56 |
| **FCLoRA** | $87.95_{\pm 0.12}$ | $95.37_{\pm 0.29}$ | $64.79_{\pm 0.20}$ | $90.76_{\pm 0.08}$ | $93.09_{\pm 0.14}$ | $83.15_{\pm 0.17}$ | $90.32_{\pm 0.86}$ | $91.22_{\pm 0.05}$ | 87.08 |

### M.2 QUESTION ANSWERING

### M.2.1 DATASET DESCRIPTION

We describe the benchmark dataset of SQuAD (Rajpurkar, 2016; Rajpurkar et al., 2018). The Stanford Question Answering Dataset (SQuAD) is a benchmark for reading comprehension, featuring questions based on Wikipedia articles. Each question is answered with a specific text segment (or span) from the corresponding passage, though some questions may have no answer at all.

- **SQuADv1.1.** [3] Over 100,000 question-answer pairs derived from more than 500 articles. The dataset consists of 87,599 samples for training and 10,570 for validation.

- **SQuADv2.0.** [4] Combines the 100,000 questions in SQuADv1.1 with over 50,000 unanswerable questions to closely resemble answerable ones. To perform well on SQuADv2.0, systems must not only provide correct answers when available but also recognize when a question cannot be answered based on the given passage and abstain from responding. The dataset consists of 130,319 samples for training and 11,873 for validation.

### M.2.2 EXPERIMENTAL SETUP

We evaluate **FCLoRA** on two question answering (QA) tasks: SQuAD v1.1 (Rajpurkar, 2016) and SQuADv2.0 (Rajpurkar et al., 2018). Following Zhang et al. (2023), we fine-tune a pre-trained DeBERTaV3$_{\text{base}}$ (He et al., 2021) with **FCLoRA** and set the rank $r$ of LoRA as $\{1, 2, 4, 8\}$. These tasks are considered as a sequence labeling problem, where the goal is to predict the probability of each token being the start and end of the answer span. We measured the performance of model using the Exact Match (EM) and F1 metrics. We use 1 GPU of NVIDIA RTX 3090 24GB for experiments. We conduct the experiments in Huggingface Framework.

### M.2.3 HYPERPARAMETERS

To tune **FCLoRA**, we fix the batch size as 16, and train 10 epochs for SQuADv1.1 and 12 epochs for SQuADv2.0, respectively. We search for the learning rate from $\{1 \times 10^{-3}, 2 \times 10^{-3}, 3 \times 10^{-3}\}$, $\bar{\sigma}$

---

[3] https://huggingface.co/datasets/rajpurkar/squad
[4] https://huggingface.co/datasets/rajpurkar/squad_v2

Table 12: Performance comparison of various methods with DeBERTaV3$_{base}$ on GLUE tasks with different random seeds. The results for the baselines are copied from (Zhang et al., 2023).

| Method | # Params | MNLI | SST-2 | CoLA | QQP | QNLI | RTE | MRPC | STS-B | Avg. |
|---|---|---|---|---|---|---|---|---|---|---|
| Full FT | 184M | 89.90 | 95.63 | 69.19 | 92.40 | 94.03 | 83.75 | 89.46 | 91.60 | 88.09 |
| BitFit | 0.10M | 89.37 | 94.84 | 66.96 | 88.41 | 92.24 | 78.70 | 87.75 | 91.35 | 86.02 |
| HAdapter | 1.22M | 90.13 | 95.53 | 68.64 | 91.91 | 94.11 | 84.48 | 89.95 | 91.48 | 88.12 |
| PAdapter | 1.18M | 90.33 | 95.61 | 68.77 | 92.04 | 94.29 | 85.20 | 89.46 | 91.54 | 88.24 |
| LoRA$_{r=8}$ | 1.33M | 90.65 | 94.95 | 69.82 | 91.99 | 93.87 | 85.20 | 89.95 | 91.60 | 88.34 |
| AdaLoRA | 1.27M | 90.76 | 96.10 | 71.45 | 92.23 | 94.55 | 88.09 | 90.69 | 91.84 | 89.31 |
| **FCLoRA** | 1.33M | 90.36$_{\pm0.03}$ | 96.33$_{\pm0.41}$ | 71.49$_{\pm0.60}$ | 92.33$_{\pm0.01}$ | 94.57$_{\pm0.05}$ | 89.41$_{\pm0.17}$ | 91.58$_{\pm1.21}$ | 92.19$_{\pm0.07}$ | 89.78 |
| HAdapter | 0.61M | 90.12 | 95.30 | 67.87 | 91.65 | 93.76 | 85.56 | 89.22 | 91.30 | 87.93 |
| PAdapter | 0.60M | 90.15 | 95.53 | 69.48 | 91.62 | 93.98 | 84.12 | 89.22 | 91.52 | 88.04 |
| HAdapter | 0.31M | 90.10 | 95.41 | 67.65 | 91.54 | 93.52 | 83.39 | 89.25 | 91.31 | 87.60 |
| PAdapter | 0.30M | 89.89 | 94.72 | 69.06 | 91.40 | 93.87 | 84.48 | 89.71 | 91.38 | 87.90 |
| LoRA$_{r=2}$ | 0.33M | 90.30 | 94.95 | 68.71 | 91.61 | 94.03 | 85.56 | 89.71 | 91.68 | 88.15 |
| AdaLoRA | 0.32M | 90.66 | 95.80 | 70.04 | 91.78 | 94.49 | 87.36 | 90.44 | 91.63 | 88.86 |
| **FCLoRA** | 0.33M | 90.66$_{\pm0.02}$ | 96.18$_{\pm0.14}$ | 71.83$_{\pm1.08}$ | 91.82$_{\pm0.07}$ | 94.50$_{\pm0.00}$ | 89.89$_{\pm1.47}$ | 91.83$_{\pm0.90}$ | 92.00$_{\pm0.23}$ | 89.83 |

Table 13: Best hyperparameters for **FCLoRA** in question answering

| Dataset | $r$ | Learning rate | $\bar{\sigma}$ | $\gamma$ |
|---|---|---|---|---|
| SQuADv1.1 | 1 | $2 \times 10^{-3}$ | $\sigma^{(2)}$ | $5 \times 10^{-1}$ |
| | 2 | $2 \times 10^{-3}$ | $\sigma^{(2)}$ | $1 \times 10^{-1}$ |
| | 4 | $1 \times 10^{-3}$ | $\sigma^{(2)}$ | $1 \times 10^{-2}$ |
| | 8 | $2 \times 10^{-3}$ | $\sigma^{(2)}$ | $1 \times 10^{-1}$ |
| SQuADv2.0 | 1 | $2 \times 10^{-3}$ | $\sigma^{(3)}$ | $7 \times 10^{-2}$ |
| | 2 | $2 \times 10^{-3}$ | $\sigma^{(3)}$ | $5 \times 10^{-2}$ |
| | 4 | $3 \times 10^{-3}$ | $\sigma^{(3)}$ | $5 \times 10^{-1}$ |
| | 8 | $3 \times 10^{-3}$ | $\sigma^{(3)}$ | $5 \times 10^{-1}$ |

from $\{\sigma^{(2)}, \sigma^{(3)}, \sigma^{(4)}\}$ and $\gamma$ from $5 \times 10^{-1}, 1 \times 10^{-1}, 7 \times 10^{-2}, 5 \times 10^{-2}, 1 \times 10^{-2}\}$. The learnable singular vectors $U/V$ are be initialized as $U$ with zeros and $V$ with random Gaussian initialization. We report the best hyperparameters of **FCLoRA** in Table 13.

## M.3 COMMONSENSE REASONING

### M.3.1 DATASET DESCRIPTION

The commonsense reasoning tasks are intended to require the model to go beyond pattern recognition. Instead, the model should use "common sense" or world knowledge to make inferences. The commonsense reasoning tasks comprise 8 sub-tasks, each with a predefined training and testing set.

- **BoolQ.** The model answers yes/no questions about short passages, testing its ability to understand statements.
- **PIQA.** The model chooses the most plausible solution for a physical interaction, focusing on practical reasoning.
- **SIQA.** The model infers the most suitable outcome or rationale in everyday social contexts.
- **HellaSwag.** The model selects the most coherent continuation of a short scenario, emphasizing commonsense inference.
- **WinoGrande.** The model resolves ambiguous pronoun references that require broad commonsense to disambiguate.
- **ARC-e.** The model tackles elementary-level science questions assessing basic scientific knowledge.

- **ARC-c.** The model addresses harder, more nuanced science questions requiring deeper reasoning.
- **OBQA.** OpenBookQA. The model answers questions using a provided 'open book' of facts, testing its ability to integrate and apply specific knowledge.

### M.3.2 EXPERIMENTAL SETUP

We follow the setting of Hu et al. (2023) and amalgamate the training datasets from all 8 tasks to create the final training dataset and conduct evaluations on the individual testing dataset for each task.

### M.3.3 HYPERPARAMETERS

To tune **FCLoRA**, the learnable singular vectors $U/V$ are initialized as $U$ with zeros, $V$ with random Gaussian initialization. We report the best hyperparameters of **FCLoRA** in Table 14.

Table 14: Best hyperparameters for **FCLoRA** in commonsense reasoning

| Model | Learning rate | Batch size | #Epochs | $\bar{\sigma}$ | $\gamma$ |
|---|---|---|---|---|---|
| LLaMA-7B | $2 \times 10^{-4}$ | 16 | 3 | $\sigma^{(2)}$ | $5 \times 10^{-1}$ |
| LLaMA2-7B | $3 \times 10^{-4}$ | 16 | 3 | $\sigma^{(4)}$ | $1 \times 10^{-2}$ |

## N FURTHER EXPERIMENTS ON MITIGATING CATASTROPHIC FORGETTING

In this section, we validate the effectiveness of **FCLoRA** on mitigating the catastrophic forgetting, following Section 5.

### N.1 MITIGATING CATASTROPHIC FORGETTING WITH OTHER MODELS

Table 15: Catastrophic forgetting across various models. 'Acc.' denotes accuracy (higher is better), and 'PPL' represents perplexity (lower is better).

| Model | RoBERTa$_{\text{MRPC}}$ | | RoBERTa$_{\text{SST-2}}$ | | LLaMA-7B | |
|---|---|---|---|---|---|---|
| | Acc.$_{\text{fine-tune}}$ | Acc.$_{\text{pre-train}}$ | Acc.$_{\text{fine-tune}}$ | Acc.$_{\text{pre-train}}$ | Acc.$_{\text{fine-tune}}$ | PPL$_{\text{pre-train}}$ |
| Pre-trained | - | 68.21 | - | 68.21 | - | 6.66 |
| LoRA | 89.05 | 18.36 | 94.81 | 54.33 | 74.7 | 8.69 |
| **FCLoRA** | **90.32** | **47.28** | **95.37** | **65.67** | **78.6** | **7.78** |

To further validate the effectiveness of **FCLoRA** on mitigating catastrophic forgetting, we report RoBERTa$_{\text{base}}$ trained on MRPC and SST-2 with accuracy for the pre-trained task with the Open-WebText (Gokaslan et al., 2019) dataset in Table 15. Furthermore, we also report the results for the large-scale model with LLaMA-7B trained on commonsense reasoning for pre-trained task with PG-19 (Rae et al., 2019). **FCLoRA** adapts to the new task with improved performance while preserving the pre-training task performance.

### N.2 CATASTROPHIC FORGETTING WITH EXPLICIT SVD-BASED LORA

To validate the catastrophic forgetting in parameterized SVD-based LoRA, we measure the accuracy on both fine-tuning and pre-training task with LoRA and its variants, including AdaLoRA (Zhang et al., 2023) and SORSA (Cao, 2024). As reported in Table 16, both AdaLoRA and SORSA exhibit significant drops in the pre-trained task accuracy after fine-tuning, with SORSA being particularly affected. In contrast, **FCLoRA** effectively mitigates catastrophic forgetting, achieving substantially higher accuracy on pre-trained tasks, while also maintaining competitive fine-tuned task performance. These results highlight the effectiveness of **FCLoRA** in retaining the pre-trained knowledge while adapting to the new task.

Table 16: Comparison of catastrophic forgetting across parameterized SVD-based LoRA methods

| Model | RoBERTa$_{\text{MRPC}}$ | | RoBERTa$_{\text{STS-B}}$ | |
|---|---|---|---|---|
| | Acc.$_{\text{fine-tune}}$ | Acc.$_{\text{pre-train}}$ | Acc.$_{\text{fine-tune}}$ | Acc.$_{\text{pre-train}}$ |
| Pre-trained | - | 61.64 | - | 61.64 |
| LoRA | 89.05 | 18.36 | 90.87 | 14.86 |
| SORSA | 89.05 | 6.45 | 90.72 | 10.37 |
| AdaLoRA | 89.05 | 25.58 | 90.62 | 28.49 |
| **FCLoRA** | **90.32** | **47.28** | **91.22** | **43.72** |

# O ADDITIONAL STUDIES

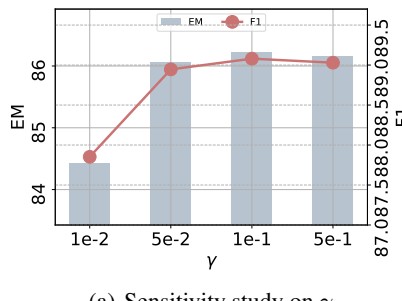

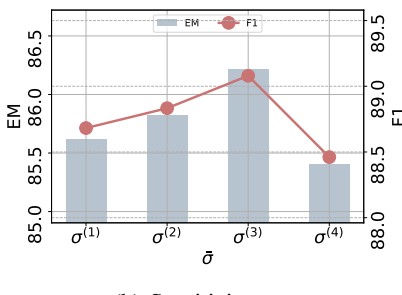

(a) Sensitivity study on $\gamma$        (b) Sensitivity on $\bar{\sigma}$

Figure 6: Sensitivity studies

## O.1 SENSITIVITY STUDY ON THE ORTHOGONAL REGULARIZATION COEFFICIENT $\gamma$

The orthogonal regularization applied to $U$ and $V$ is used to learn the singular values that consists the injected fine-grained singular components. We further conduct sensitivity study on the effect of the orthogonal regularization coefficient $\gamma$. We fine-tuned the DeBERTaV3$_{\text{base}}$ model on the SQuADv2.0 dataset. As shown in Fig. 6 (a), appropriate regularization induces the orthogonalization of singular values, leading to improved convergence during fine-tuning and enhanced performance. However, excessive regularization results in performance degradation, indicating the need for an optimal balance that maximizes the benefits of regularization without hindering the ability of model to learn task-specific patterns.

## O.2 SENSITIVITY ON THE UPPER BOUND OF INJECTED SINGULAR VALUE $\bar{\sigma}$

We constrain the maximum value of the parameterized singular values with the hyperparameter $\bar{\sigma}$ to learn the injected fine-grained singular components. To analyze the impact of $\bar{\sigma}$ on performance, we fine-tune the DeBERTaV3$_{\text{base}}$ model on the SQuADv2.0 dataset and report EM/F1 score according to $\bar{\sigma}$. In our experiments, $\bar{\sigma}$ holds the $q$-th quantile value of the singular values distribution of $W_0$, denoted as $\sigma^{(q)}$. As illustrated in Fig. 6 (b), the performance peaks when $\bar{\sigma} = \sigma^{(3)}$, indicating that $\bar{\sigma}$ at the appropriately small value level allows the model to optimally learn the injected fine-grained singular components while maintaining best accuracy. However, reducing or increasing $\bar{\sigma}$ too much leads to a degradation in both EM and F1 scores, suggesting that an inappropriate scale disrupts the capability of model to learn fine-grained details effectively.

## O.3 SENSITIVITY ON THE RANK $r$ ON CATASTROPHIC FORGETTING

To verify whether **FCLoRA** maintains its performance and continues to mitigate forgetting as the rank increases, we conduct sensitivity study on the rank $r$ on MRPC and STS-B dataset. Specifically, we measured the largest singular value and Frobenius norm of the difference between the pre-trained

model and the fine-tuned model. Also, we evaluated the accuracy and the evaluation loss on the pre-trained task.

Table 17: The effect of rank $r$ on catastrophic forgetting

| $r$ | Metric | 8 | 16 | 64 |
|---|---|---|---|---|
| | Spectral norm | 0.94 | 0.70 | 0.36 |
| MRPC | Eval. $\text{loss}_{\text{pre-train}}$ | 4.35 | 4.18 | 3.20 |
| | $\text{Acc.}_{\text{pre-train}}$ | 32.00 | 33.58 | 41.32 |
| | Spectral norm | 0.94 | 1.40 | 1.12 |
| STSB | Eval. $\text{loss}_{\text{pre-train}}$ | 3.18 | 2.49 | 2.53 |
| | $\text{Acc.}_{\text{pre-train}}$ | 43.72 | 51.15 | 48.79 |

As reported in Table 17, as $r$ changes, the largest singular value also varies, which, in turn affects the performance on the pre-trained task. The performance on the pre-trained task, however, does not degrade but rather shows an improvement. This indicates that the proposed model retains its ability to effectively mitigate catastrophic forgetting even as the rank increases.

### O.4 ORTHOGONAL REGULARIZATION ON PARAMETERIZED SINGULAR VECTORS

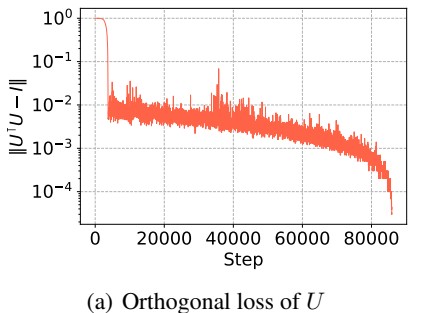
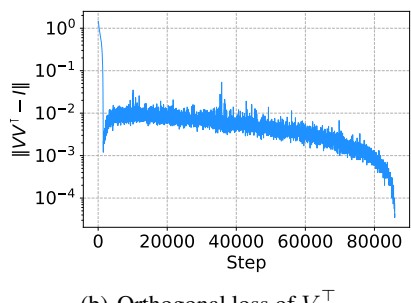

(a) Orthogonal loss of $U$         (b) Orthogonal loss of $V^{\top}$

Figure 7: The orthogonal loss curves of parameterized singular vectors $U$ and $V$ when fine-tuning $\text{RoBERTa}_{\text{base}}$ on STS-B dataset

Fig. 7 shows the orthogonal loss curve of parameter singular vectors $U$ and $V$ of $\text{RoBERTa}_{\text{base}}$ fine-tuned on STS-B dataset. The singular vectors are orthogonally optimized as indicated by the consistent reduction in orthogonal loss.

### O.5 GRAM-SCHMIDT ORTHOGONALIZATION ON PARAMETERIZED SINGULAR VECTORS

Table 18: Comparison of Gram-Schmidt orthogonalization and orthogonal regularization on various datasets. Metrics are reported as "accuracy (%) ↑ / minutes per epoch ↓".

| | CoLA | RTE | MRPC | STS-B |
|---|---|---|---|---|
| Gram-Schmidt ortho. | $57.31_{\pm 0.85}/3.6$ | $67.27_{\pm 1.23}/1.1$ | $87.58_{\pm 1.02}/2.0$ | $89.88_{\pm 0.19}/2.6$ |
| Ortho. Regularization | $\mathbf{64.79}_{\pm 0.20}/\mathbf{2.9}$ | $\mathbf{83.15}_{\pm 0.17}/\mathbf{0.8}$ | $\mathbf{90.32}_{\pm 0.86}/\mathbf{1.3}$ | $\mathbf{91.22}_{\pm 0.05}/\mathbf{2.1}$ |

Orthogonal regularization is a widely used method for implementing parameterized SVD in LoRA (Zhang et al., 2023; Cao, 2024; Zhang et al., 2024). Unlike orthogonal regularization, Gram-Schmidt orthogonalization—one of the orthogonalization methods—produces strictly orthogonal vectors. However, since it refines each subsequent vector based on the previously orthogonalized ones, it has the disadvantage of being difficult to parallelize compared to the regularization approach. Furthermore, as shown in Fig. 7, the orthogonal error rapidly decreases to below $10^{-2}$ in the early stages of fine-tuning, indicating that the singular vectors can be sufficiently trained. We

conduct an ablation study with orthogonal regularization by replacing it with Gram-Schmidt-based orthogonalization. We measured the performance and training speed in Table 18.

## O.6 TRAINING STABILITY OF FCLoRA

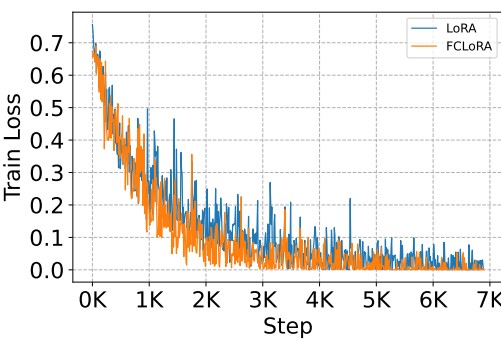

Figure 8: Loss curves on the MRPC dataset on RoBERTa$_{\text{base}}$ comparing LoRA and FCLoRA

To evaluate the training stability of our method, we compare the loss curves of LoRA and our proposed **FCLoRA** on the MRPC dataset with RoBERTa$_{\text{base}}$. Although **FCLoRA** incorporates an additional orthogonal regularization term, the overall training trajectory demonstrates significantly improved stability. As shown in Figure 8, FCLoRA shows smoother and faster stabilization, which indicates that the orthogonality constraint on the parameterized singular vectors and clipping the singular values not only enhances representational robustness but also leads to more stable optimization behavior.

## P  COMPARISON OF COMPUTATIONAL COMPLEXITY

**Increased number of trainable parameters.** For the hidden dimension $d$ and the rank of adapter $r$, conventional LoRA learns $2dr$ parameters per layer. **FCLoRA** introduces only an additional $r$ parameter to the existing LoRA, resulting in a total of $2dr + r$ parameters per layer. For instance, when $r = 2$ as shown in Table 10, conventional LoRA learns 331,776 parameters, whereas **FCLoRA** learns a total of 331,920 parameters—the additional parameters are only 144.

**Increased computational cost.** In inference stage, conventional LoRA involves computations with complexity $\mathcal{O}(d^2 r)$ per layer, whereas **FCLoRA** includes the process of multiplying the parameterized singular value with the parameterized singular vector, resulting in an additional complexity of $\mathcal{O}(d^2 r + dr)$, which is comparable. Note that the explicit SVD is performed only once during the initialization stage and is not required during fine-tuning.

Table 19: Comparison of training time (min per epoch) and peak GPU usage (GB)

| Method | MNLI | SST-2 | CoLA | QQP | QNLI | RTE | MRPC | STS-B |
|---|---|---|---|---|---|---|---|---|
| LoRA | 105.9/24.9 | 18.2/24.9 | 2.3/24.9 | 98.1/24.9 | 28.3/24.9 | 0.7/24.9 | 1.0/12.5 | 1.6/24.9 |
| PiSSA | 106.2/24.9 | 18.1/24.9 | 2.3/24.9 | 98.1/24.9 | 28.2/24.9 | 0.7/24.9 | 1.0/12.5 | 1.5/24.9 |
| AdaLoRA | 123.4/25.6 | 21.1/25.6 | 2.7/25.6 | 114.4/25.6 | 33.1/25.6 | 0.8/25.6 | 1.3/13.1 | 1.8/25.6 |
| MiLoRA | 106.0/24.9 | 18.1/24.9 | 2.3/24.9 | 98.1/24.9 | 28.2/24.9 | 0.7/24.9 | 1.0/12.5 | 1.5/24.9 |
| **FCLoRA** | 128.9/25.2 | 22.1/25.2 | 2.8/25.2 | 119.4/25.2 | 34.3/25.2 | 0.8/25.2 | 1.3/12.8 | 1.9/25.2 |

**Empirical complexity analysis.** We conduct additional experiments on empirical time complexity and GPU usage during fine-tuning and inference phase. Table 19 summarizes the empirical training time (min per epoch) and peak GPU usage (GB) of RoBERTa$_{\text{base}}$ fine-tuned on GLUE tasks. The GPU usage showed a very slight increase compared to the original LoRA, and the additional runtime occurs in **FCLoRA** and AdaLoRA. This increase arises from the orthogonal regularization

Table 20: Comparison of inference time (min per epoch) and peak GPU usage (GB)

| Method | MNLI | SST-2 | CoLA | QQP | QNLI | RTE | MRPC | STS-B |
|---|---|---|---|---|---|---|---|---|
| LoRA | 0.85/0.3 | 0.08/0.3 | 0.10/0.3 | 3.46/0.3 | 0.47/0.3 | 0.03/0.3 | 0.05/0.3 | 0.14/0.3 |
| PiSSA | 0.84/0.3 | 0.08/0.3 | 0.09/0.3 | 3.47/0.3 | 0.47/0.3 | 0.03/0.3 | 0.04/0.3 | 0.13/0.3 |
| AdaLoRA | 1.32/0.3 | 0.13/0.3 | 0.15/0.3 | 5.43/0.3 | 0.74/0.3 | 0.05/0.3 | 0.07/0.3 | 0.21/0.3 |
| MiLoRA | 0.84/0.3 | 0.08/0.3 | 0.09/0.3 | 3.47/0.3 | 0.47/0.3 | 0.03/0.3 | 0.04/0.3 | 0.13/0.3 |
| **FCLoRA** | 1.01/0.3 | 0.10/0.3 | 0.12/0.3 | 4.15/0.3 | 0.57/0.3 | 0.04/0.3 | 0.05/0.3 | 0.16/0.3 |

of singular vectors generated by parameterized SVD. However, fine-tuning typically requires fewer epochs, and considering the improved performance and the ability to retain pre-trained knowledge compared to the baseline model, this increase is negligible. Table 20 reports the empirical inference time (min per epoch) and peak GPU usage (GB) of RoBERTa$_{base}$ fine-tuned on GLUE tasks. **FCLoRA** exhibits a marginal increase in inference latency and peak GPU memory relative to vanilla LoRA; nonetheless, both metrics remain lower than those of AdaLoRA and are practically comparable overall.

