# OpenReview forum: "FCLoRA: Low-Rank Adaptation with Fine-Grained Component Injection"
_ICLR.cc/2026/Conference — ICLR 2026 Conference Withdrawn Submission_

### Official Review · Reviewer_5YNq · 2025-10-23

**Soundness:** 2
**Presentation:** 3
**Contribution:** 2
**Rating:** 4
**Confidence:** 3

**Summary:**

The paper proposes FCLoRA, a parameter-efficient fine-tuning framework that enhances LoRA by injecting fine-grained singular components into pretrained models using a parameterized SVD formulation.  The authors analyze how different singular value ranges in pretrained weights correspond to transferable versus task-specific knowledge.

**Strengths:**

1. The paper provides a principled analysis of the relationship between singular value spectra, task transferability, and forgetting, offering new geometric insight into PEFT.
2. The parameterized SVD with bounded singular values is simple yet effective, easily integrable into existing LoRA pipelines.
3. Directly mitigates catastrophic forgetting — a well-known but underexplored problem in PEFT research.

**Weaknesses:**

1. I found that the conclusions on the principle subspace are somehow contradict to those in PISSA. Can the author explain the reason?
2. The paper introduces an upper bound on singular values to prevent catastrophic forgetting. However, LoRA’s empirical success partially relies on allowing singular value amplification for better expressivity. How do you justify that a static bound does not suppress task-specific adaptation capacity, especially for highly anisotropic pretrained representations?
3. How does FCLoRA differ in principle from AdaLoRA or PiSSA, which also modulate the singular spectrum during training? Is the key contribution the explicit upper bound, or does the fine-grained injection scheme lead to qualitatively different subspace evolution?
4. Since SVD is required for each adapter, how does the computational overhead scale with model size? Are there efficient approximations to make FCLoRA practical for industrial-scale models?
5. Could the authors provide quantitative evidence that this stabilization directly correlates with reduced forgetting, beyond task accuracy—e.g., via spectral entropy or principal angle evolution?

**Questions:**

Please see weaknesses

---

> ### Author Response · Authors · 2025-11-21
> **Authors Response (1/3)**
>
> W1. The conclusions on the principle subspace are somehow contradict to those in PISSA.
>
> A1. Our conclusion is not in conflict with PiSSA. Both PiSSA and MiLoRA are built upon the same spectral assumption: the principal components of the pre-trained weights encode essential information, whereas the minor components capture finer-grained and noisier variations. Based on this assumption, PiSSA argues that the principal subspace should be directly updated because it contains important information, while MiLoRA takes the complementary view that this information should instead be preserved, updating only the minor components.
>
> FCLoRA moves beyond this perspective of 'information importance' and is instead motivated by the observation that the degree of alignment between the pre-training task and the fine-tuning task varies across components. The principal directions encode global world knowledge and are already strongly aligned across the two tasks, and thus require little modification. In contrast, the minor directions exhibit low alignment and are therefore more suitable for injecting task-specific information.
>
> Thus, our approach does not contradict PiSSA; rather, it extends the principal–minor interpretation by emphasizing that the decomposition reflects not only information importance but also the alignment structure between pre-training and fine-tuning. In other words, while PiSSA posits that “principal components contain important information and should be directly updated,” we empirically show—via Fisher overlap analysis—that these principal directions primarily encode shared knowledge between the tasks and therefore do not require substantial alteration. Conversely, the fine-grained components, which exhibit low task alignment, are the most appropriate subspace for incorporating new information.
>
> Viewed from this perspective, FCLoRA does not oppose PiSSA or MiLoRA. Instead, it integrates and refines the assumptions and empirical findings of both methods at a more granular level:
> - PiSSA’s emphasis on the importance of principal components naturally aligns with our interpretation that these directions correspond to pre-trained shared knowledge that should be preserved, and
> - MiLoRA’s view that minor components should be updated is empirically supported by our alignment-based analysis (Figure 1 (a)).
>
> However, both PiSSA and MiLoRA primarily consider initialization and do not account for the potential information loss caused by direct modification of the principal subspace or for the evolution of the spectrum during fine-tuning—factors that can contribute to catastrophic forgetting.
>
> In summary, FCLoRA does not conflict with existing methods. Rather, by reinterpreting the spectral structure in terms of task alignment rather than information importance, FCLoRA provides a more general and coherent explanation of which components should be updated and why in order to minimize catastrophic forgetting while maximizing task adaptability.

---

> ### Author Response · Authors · 2025-11-21
> **Authors Response (2/3)**
>
> W2. How do you justify that a static bound does not suppress task-specific adaptation capacity, especially for highly anisotropic pretrained representations?
>
> [FCLoRA focus capacity on useful representations rather than restricting it]
>
> The strength of a pre-trained LM largely stems from its ability to encode global world knowledge learned from a massive corpus. In many open-source LLMs, the pre-training stage is also understood as the phase during which the model acquires broad, transferable knowledge useful for downstream tasks. Therefore, the objective of fine-tuning—particularly in PEFT—is not to “re-learn” new knowledge from scratch, but rather to reuse and adapt this pre-trained global knowledge efficiently within a limited parameter budget.
>
> As in Figure 1, principal components associated with large singular values exhibit high Fisher overlap between the pre-training and fine-tuning tasks and are thus already well aligned. In contrast, components associated with small singular values tend to be fine-grained and task-specific, requiring greater adaptation. From this perspective, the spectral clipping in FCLoRA should not be viewed as a constraint that reduces overall capacity, but rather as a mechanism that allocates the limited parameter budget more effectively between the aligned (global) and task-specific (fine-grained) components.
>
> FCLoRA preserves the dominant components to maintain pre-trained knowledge, while injecting fine-grained information only within a controlled, clipped singular value range. This encourages the model to selectively learn new directions (unaligned singular vectors) absent in the pre-trained weight. Such clipping acts as a regularizer that prevents the representation space from expanding excessively in undesirable directions and, from a MAP viewpoint, helps preserve the pre-training prior, thereby reducing catastrophic forgetting.
>
> In Appendix L, FCLoRA not only acquires fine-grained information absent from the pre-trained weights but also maintains pre-trained task performance more effectively than existing LoRA variants. This supports that the performance gains arise not from hyperparameter variance, but from the proposed spectral constraint’s ability to focus capacity on useful representations rather than restricting it.
>
> [Singular Value Clipping in other domains]
>
> Moreover, singular value clipping (SVP), which is closely related to our approach, has been widely used across diverse domains such as GANs and CNNs. For example, [1] introduces a K-Lipschitz constraint to overcome instability in GANs and achieves it via SVP. In [2], the authors show that the singular values of convolutional layers are key contributors to exploding and vanishing gradients; by controlling the Lipschitz constant through SVP, they prevent these issues and improve generalization performance. Additionally, appropriate clipping has been shown to enhance network stability and test-time generalization. Similarly, [3] demonstrates that regularizing per-layer spectral norms improves both model generalization and adversarial robustness. Even though FCLoRA does not adopt these SVP formulations verbatim, the underlying principle suggests that comparable effects—such as enhanced stability and better-conditioned updates.
>
> [Additional clarification on anisotropic pretrained representations]
>
> Importantly, highly anisotropic pretrained representations do not contradict our approach; rather, they further motivate the need for controlled spectral behavior. In anisotropic LMs, the dominant singular directions encode global world knowledge and exhibit inherently high alignment with a wide range of downstream tasks, as also confirmed by our Fisher-overlap analysis (Figure 1 (a)). These directions therefore require little modification for effective adaptation, and clipping them does not suppress expressivity.
>
> In contrast, task-specific information primarily resides in the low-singular, low-alignment subspace. FCLoRA allocates adaptation capacity exactly to this region, enabling the model to learn new directions without perturbing the anisotropic structure that carries useful pretrained knowledge. Notably, in highly anisotropic settings, uncontrolled amplification of dominant singular directions can cause severe representation drift and exacerbate catastrophic forgetting—an effect we empirically observe in existing LoRA variants. By preventing such misaligned amplification while preserving the pretrained anisotropy, FCLoRA maintains stability and encourages adaptation in the most relevant subspace.
>
> Thus, even in strongly anisotropic pretrained models, a static bound does not restrict adaptation capacity; it instead ensures that expressivity is exercised in task-aligned directions rather than in highly dominant but already well-aligned components.

---

> ### Author Response · Authors · 2025-11-21
> **Authors Response (3/3)**
>
> W3. How does FCLoRA differ in principle from AdaLoRA or PiSSA, which also modulate the singular spectrum during training?
>
> - We kindly refer the reviewer to the General Response section, 'Novelty of FCLoRA compard to SVD-based model'.
>
> [Subspace analysis between $W$ and $\Delta W$]
>
> In Appendix L, we have performed a projection-based analysis, following the methodology introduced in the original LoRA paper, to quantitatively examine the relationship between the subspaces learned by the pre-trained weights and the adapter. To measure this relationship, we compute the following two quantities:
>
> 1. ${Factor}_{W \rightarrow \Delta W}$ — how much the adapter amplifies pre-trained information
> 2. ${Factor}_{\Delta W \rightarrow  W}$  — how much new information is encoded in the adapter
>
> To address the reviewer’s concern, we define the following ratio based on these two values:
>
> $\text{Ratio} = \frac{\text{Factor}_{\Delta W \rightarrow W}}{\text{Factor}_{W \rightarrow \Delta W}}$
>
> A higher Ratio indicates that \(\Delta W\) contains new components that do not lie within the subspace of the pre-trained weights.
>
> In Appendix L, the value computed for FCLoRA is:
>
> $\frac{31.18}{2.60} = 11.99$
>
> This value is significantly higher than those of existing LoRA-based methods, demonstrating that FCLoRA is not merely amplifying existing directions, but instead learning new task-specific directions that do not exist in the pre-trained weights.
>
> In summary, FCLoRA preserves pre-trained knowledge while injecting fine-grained information into low-alignment regions, thereby naturally inducing a divergence of subspaces during fine-tuning. All analysis details and numerical results are thoroughly documented in Appendix L.
>
> W4. Since SVD is required for each adapter, how does the computational overhead scale with model size? Are there efficient approximations to make FCLoRA practical for industrial-scale models?
>
> - The initialization needs to be performed only once, which is a common strategy in many Explicit-SVD–based models.
> - As a practical method, for example, the layers can be divided into several consecutive segments, and a single representative layer can be selected from each segment to compute its SVD. The resulting spectral parameters (e.g., $\bar{\sigma}$) are then propagated to the remaining layers within the same segment. For example, layers 0–4 may share the $\bar{\sigma}$ obtained from layer 0, while layers 5–9 use the $\bar{\sigma}$ computed from layer 5 and apply it to layers 6–9, and the remaining segments can be treated in the same manner. This segment-wise spectral sharing significantly reduces the initialization cost without requiring SVD computation for every layer.
>
>
> W5. Could the authors provide quantitative evidence that this stabilization directly correlates with reduced forgetting, beyond task accuracy—e.g., via spectral entropy or principal angle evolution?
>
> As shown in Theorem 3.1, the direct cause of catastrophic forgetting is the erosion of the pre-trained spectrum caused by the adapter's singular values interfering with the dominant singular components of the base model. FCLoRA is explicitly designed to avoid this issue: the principal components of the pre-trained weight are preserved, while task-specific information in non-aligned, fine-grained directions is encoded only in minor singular components. Consequently, it is natural for the update direction to be less aligned with the original weight, leading to a larger principal angle between $W$ and $\Delta W$. This geometric deviation is an *intended* behavior rather than a sign of forgetting.
>
> Furthermore, FCLoRA constrains the singular values with an upper bound, effectively preventing spectral explosion. Therefore, a higher spectral entropy does not imply erosion of dominant directions; instead, it reflects that the update energy is distributed across multiple fine-grained directions within a normalized and bounded spectral range.
>
> This interpretation is also consistent with our measurements:
>
> | Model  | Spectral entropy | Principal angle |
> |--------|---|------|
> | LoRA   | 0.1190           | 1.4489           |
> | PiSSA  | 0.0927           | 1.4475           |
> | MiLoRA | 0.0688           | 1.4433           |
> | FCLoRA | **0.3125**         | **1.4829**         |
>
> Although FCLoRA shows the highest spectral entropy and principal angle, these values do not indicate catastrophic forgetting. Instead, they arise naturally from preserving the principal subspace of the pre-trained model while expanding expressivity along fine-grained, non-aligned directions—which is precisely the design objective of FCLoRA.
>
> Additionally, for a performance-based confirmation of catastrophic forgetting, please refer to the General Response 'Mitigation of Catastrophic Forgetting', where we additionally show that FCLoRA maintains pre-trained capabilities more effectively than prior methods.

---

> ### Author Response · Authors · 2025-11-21
> **References**
>
> > [1] Saito, Masaki, Eiichi Matsumoto, and Shunta Saito. "Temporal generative adversarial nets with singular value clipping." Proceedings of the IEEE international conference on computer vision. 2017.
>
> > [2] Senderovich, Alexandra, et al. "Towards practical control of singular values of convolutional layers." Advances in Neural Information Processing Systems 35 (2022): 10918-10930.
>
> > [3] Boroojeny, Ali Ebrahimpour, Matus Telgarsky, and Hari Sundaram. "Spectrum extraction and clipping for implicitly linear layers." International Conference on Artificial Intelligence and Statistics. PMLR, 2024.

---

### Official Review · Reviewer_54EY · 2025-10-27

**Soundness:** 2
**Presentation:** 2
**Contribution:** 2
**Rating:** 2
**Confidence:** 4

**Summary:**

This paper proposes FCLoRA, a modification of the Low-Rank Adaptation (LoRA) method for parameter-efficient fine-tuning. The motivation stems from two main observations derived from Singular Value Decomposition (SVD) analysis of network parameters: 1) Dominant singular components (large singular values) of pre-trained weights are reusable, while fine-grained components (small singular values) are more task-specific and need adaptation. 2) The growth of singular values in the LoRA adapter ($\Delta W$) during fine-tuning can lead to catastrophic forgetting of pre-trained knowledge, linked theoretically to a decrease in the prior probability term in MAP estimation ($p (\theta|\mathcal{D}_A) $). FCLoRA addresses these by injecting a learnable low-rank update $\Delta W$ formulated using parameterized SVD ($\Delta W = U\Sigma V^\top$). Critically, it restricts the learnable singular values $\sigma_n \in \Sigma$ to be within an "appropriate range" by clipping them below a pre-defined upper bound $\bar{\sigma}$ (e.g., a quantile $\sigma^{(q)}$ of $W_0$'s singular values). This aims to focus learning on the fine-grained domain and prevent excessive singular value growth, thereby improving adaptation while mitigating forgetting. Experiments on NLU, QA, and commonsense reasoning tasks are presented.

**Strengths:**

1. The paper provides a reasonable motivation based on SVD analysis, distinguishing the roles of dominant vs. fine-grained components and linking singular value growth in adapters to catastrophic forgetting via a theoretical argument (Theorem 3.1). Figure 1 visually supports these claims.

2. The core idea of using parameterized SVD and explicitly constraining the magnitude of learnable singular values ($\Sigma$) with an upper bound $\bar{\sigma}$ is a direct and intuitive way to implement the paper's motivation of focusing on "fine-grained" adaptation and preventing excessive deviation from the pre-trained state.

**Weaknesses:**

1. Limited Novelty and Distinction from Prior Work: The use of parameterized SVD in LoRA variants is not new (e.g., AdaLoRA). Several recent works also leverage SVD components for LoRA initialization or modification, such as PISSA, MiLoRA, LoRA-XS, and SORSA. FCLoRA's specific contribution lies in clipping the learnable singular values $\Sigma$ to an upper bound $\bar{\sigma}$. The paper fails to sufficiently articulate why this specific mechanism is significantly novel or superior compared to these closely related methods (e.g., MiLoRA which also focuses on smaller components, or AdaLoRA which prunes components). Experimental comparisons against PISSA and MiLoRA are notably absent despite their relevance.


2. Crucial Hyperparameter $\bar{\sigma}$ Underexplored: The choice of the upper bound $\bar{\sigma}$ is critical to FCLoRA's definition of "fine-grained" injection. How should $\bar{\sigma}$ be chosen? The paper suggests using a quantile $\sigma^{(q)}$ of $W_0$'s singular values but provides little guidance or analysis on which quantile works best, how sensitive the results are to this choice, or whether it needs tuning per task/model. The ablation study in Appendix O.2 shows sensitivity but lacks a principled selection method. This ambiguity makes the method difficult to apply reliably.

3. Insufficient Validation of Forgetting Mitigation: The primary claim regarding catastrophic forgetting mitigation relies heavily on Theorem 3.1 (based on Laplace approximation) and experiments measuring performance degradation on the pre-training task after single-task fine-tuning (Section 5.3, Figure 3). While indicative, this does not adequately simulate or evaluate performance in a proper continual learning setting with a sequence of distinct downstream tasks. Standard CL benchmarks and metrics (like backward transfer or average accuracy over a sequence) are needed to robustly validate the claims about mitigating forgetting in practice.

4. Computational Aspects: Parameterized SVD requires learning $U, \Sigma, V$ and potentially enforcing orthogonality via regularization $R(U,V)$ with coefficient $\gamma$. How does the computational cost and optimization stability compare to standard LoRA (learning $A, B$)? The paper claims negligible overhead but provides limited analysis (Appendix P shows increased training/inference time).

**Questions:**

1. Can the authors clearly differentiate FCLoRA's contribution from related works like PISSA, MiLoRA, AdaLoRA, and SORSA, both conceptually and empirically (e.g., through direct experimental comparisons)?

2. How should the crucial hyperparameter $\bar{\sigma}$ be set in practice? Can the authors provide guidelines or further sensitivity analysis across different models and task types? Is the quantile approach generally applicable?

3. To substantiate the claim of mitigating catastrophic forgetting, could the authors evaluate FCLoRA on standard continual learning benchmarks involving task sequences and report metrics like average accuracy and backward transfer, comparing against relevant CL baselines?

4. Can the authors provide more details on the practical training stability and convergence speed of learning parameterized SVD components ($U, \Sigma, V$ with orthogonality constraints) compared to standard LoRA ($A, B$)?

---

> ### Author Response · Authors · 2025-11-21
> **Authors Response (1/2)**
>
> W1. Limited Novelty and Distinction from Prior Work.
>
> - We kindly refer the reviewer to the General Response section, 'Novelty of FCLoRA compard to SVD-based model'.
>
> W2. Crucial Hyperparameter $\bar{\sigma}$ Underexplored.
>
> -  We kindly refer the reviewer to the General Response section, 'Detailed explanation of $\bar{\sigma}$'.
>
> W3. Insufficient Validation of Forgetting Mitigation: Standard CL benchmarks and metrics (like backward transfer or average accuracy over a sequence) are needed to robustly validate the claims about mitigating forgetting in practice.
>
> - We kindly refer the reviewer to the General Response section, 'Mitigation of Catastrophic Forgetting.'
>
> W4. Computational Aspects: Parameterized SVD requires learning $U, \Sigma, V$ and potentially enforcing orthogonality via regularization $R(U, V)$ with coefficient $\gamma$. How does the computational cost and optimization stability compare to standard LoRA (learning $A, B$)? The paper claims negligible overhead but provides limited analysis.
>
> - In Appendix P, we have already reported the increase in parameter count, the analytical computational cost, and the comparison of training and inference time across LoRA, explicit SVD-based LoRA, and parameterized SVD–based LoRA variants. As demonstrated in these analyses, our method retains computational requirements that are fully comparable to the existing LoRA family.
>
> Q1. Can the authors clearly differentiate FCLoRA's contribution from related works like PISSA, MiLoRA, AdaLoRA, and SORSA, both conceptually and empirically (e.g., through direct experimental comparisons)?
>
> - We kindly refer the reviewer to the General Response section, 'Novelty of FCLoRA compard to SVD-based model'.
>
> Q2. How should the crucial hyperparameter $\bar{\sigma}$ be set in practice? Can the authors provide guidelines or further sensitivity analysis across different models and task types? Is the quantile approach generally applicable?
>
> A2.
> We conducted a grid search, and the corresponding search ranges as well as the optimal hyperparameters are thoroughly reported in the appendix M. In addition, we performed a sensitivity study to further analyze the behavior of $\bar{\sigma}$. For more detailed explanations and results, we kindly refer the reviewer to the General Response, 'Detailed Explanation of $\bar{\sigma}$'.

---

> ### Author Response · Authors · 2025-11-21
> **Authors Response (2/2)**
>
> Q3. To substantiate the claim of mitigating catastrophic forgetting, could the authors evaluate FCLoRA on standard continual learning benchmarks involving task sequences and report metrics like average accuracy and backward transfer, comparing against relevant CL baselines?
>
> A3.
> - Continual learning with order: SST-2 -> CoLA -> RTE -> MRPC
>
> | Model     | SST-2$_{finetune}$ | SST-2$_{pretrain}$ | CoLA$_{finetune}$ | CoLA$_{pretrain}$ | RTE$_{finetune}$ | RTE$_{pretrain}$ | MRPC$_{finetune}$ | MRPC$_{pretrain}$ |
> |--|--|--|--|---|--|--|---|---|
> | Pre-train | - | 61.64 | - | 61.64 | - | 61.64   | - | 61.64    |
> | LoRA      | 94.50 | 45.15 | 59.74     | 48.51    | 71.00    | 40.47   | 88.97     | 33.38    |
> | MiLoRA    | 94.76 | 48.64 | 59.77     | 52.64    | 71.00    | 47.33   | 88.24     | 29.77    |
> | FCLoRA    | **95.37**  | **51.76** | **60.77** | **54.04**| **79.18**| **52.44**| **89.30** | **45.08**|
>
> Although our main paper focuses on single-step fine-tuning, we additionally investigate the effectiveness of our method in a continual learning setting where more than two tasks are learned sequentially. Specifically, we conduct four consecutive fine-tuning rounds on GLUE tasks. In this setup, the pre-trained weights are kept frozen, and only the LoRA weights are loaded from the previous task and further updated. We observe that our method exhibits a noticeably lower rate of forgetting compared to LoRA and MiLoRA.
>
> - Continual learning benchmark from [1], AA ($\uparrow$), BWT ($\uparrow$)
>
> | Model  | Order-1 | Order-2 | Order-3 | avg |
> |---| -|--|---|--|
> | O-LoRA | 66.28 / -14.91  | 68.35 / -12.54 | 71.06 / -7.80   | 68.75 / -11.75 |
> | FCLoRA | **75.75 / -4.80** | **77.32 / -2.14** | **77.12 / -2.40** | **76.73 / -3.11** |
>
> To evaluate the performance of our model under a standard continual learning benchmark, we follow the experimental framework of O-LoRA [1], which is LoRA-based method for continual learning. For O-LoRA, we directly use the original implementation released on GitHub, and we measure both Average Accuracy (AA) and Backward Transfer (BWT). Following the original paper, AA is computed as the mean accuracy over all tasks when evaluated after completing the last task. For both metrics, higher is better.
>
> More detailed explanations and additional results are provided in the General Response 'Mitigation of Catastrophic Forgetting'.
>
> Q4. Can the authors provide more details on the practical training stability and convergence speed of learning parameterized SVD components ($U,\Sigma,V$ with orthogonality constraints) compared to standard LoRA ($A,B$)?
>
> Although LoRA does not impose orthogonality constraints and thus cannot be directly compared, we include an analysis of the convergence behavior of singular vectors in the Appendix O.4. We will aslo provide an additional loss plot in Appendix of revised paper. Despite the fact that the loss includes both the task loss and the orthogonal regularization loss, we observe that FCLoRA converges faster and more stably than the standard LoRA.
>
>
> ---
>
> > [1] Wang, Xiao, et al. "Orthogonal subspace learning for language model continual learning." Findings of the Association for Computational Linguistics: EMNLP 2023. 2023.

---

### Official Review · Reviewer_KCpv · 2025-10-30

**Soundness:** 2
**Presentation:** 2
**Contribution:** 2
**Rating:** 2
**Confidence:** 4

**Summary:**

This paper proposes FCLoRA, a LoRA variant that uses a parameterized SVD of the adapter and focuses the learnable part on fine-grained singular components, while keeping dominant components within a bounded range. The motivation has two parts. First, an empirical SVD analysis suggests that large singular components of pre-trained weights can often be reused across downstream tasks, while smaller components are more task-specific. Second, during fine-tuning, the singular values of a LoRA adapter can grow without control, and the authors link this growth to catastrophic forgetting via a MAP style argument. FCLoRA injects a low rank update through SVD and clips the learnable singular values using a quantile of the spectrum of the pre-trained weight. The goal is to let the model adapt to new tasks without erasing the pre-trained prior. Experiments on NLU, QA and several reasoning tasks show small but mostly consistent gains over common LoRA baselines.

**Strengths:**

I like the attempt to tell a fully spectral story. The paper explains that dominant components can be reused, that fine grained components need to be adapted, and that uncontrolled growth of singular values weakens the pre-trained prior. This is intuitive and the figures support it. The method itself is conceptually light, it is just LoRA with an SVD front and a clipping rule for the singular values. The experiments cover several models and tasks, which shows that the idea is at least practical.

**Weaknesses:**

The main problem is limited novelty compared with recent SVD based LoRA work. Learning U, Sigma and V for LoRA, or choosing and bounding the singular values, is already present in several papers. This work does not fully isolate what FCLoRA adds beyond saying that the goal is to mitigate forgetting.

A second problem is that the crucial hyperparameter, the quantile of the pre-trained spectrum that defines the upper bound on Sigma, is under explained. The paper simply says to use a quantile of the singular values, but it does not show how sensitive the results are to this choice, whether different models or tasks need different quantiles, or whether per layer and global quantiles behave differently. At the moment this looks like a hand tuned knob.

A third problem is that the evidence for forgetting is weak. The central story of the paper is catastrophic forgetting, but the experiments mainly check how much performance on the pre-training task is lost after fine-tuning a single task. This is not the same as showing mitigation on a real continual learning sequence with several distinct downstream tasks and with standard CL metrics such as average accuracy and backward transfer. Without this, it is hard to give credit for the claimed mitigation.

Finally, the computational aspect of learning a parameterized SVD, including possible orthogonality regularization, is not quantified. There are no numbers for wall clock time, peak memory, or extra parameters, compared to standard LoRA or to another SVD style baseline.

**Questions:**

1. Can you run at least one sequential or continual experiment, for example a chain of four or five NLU tasks, and show that FCLoRA forgets slower than standard LoRA and at least one recent SVD LoRA baseline?
2. Can you provide a sensitivity analysis of the clipping quantile, for example per layer versus global, and quantiles such as 0.6, 0.7, 0.8 and 0.9, on two different model and task pairs?
3. Can you report training time, peak memory and number of extra parameters for FCLoRA, for standard LoRA, and for one SVD LoRA baseline, on the same hardware and with the same sequence length?
4. Can you clarify in a short ablation in which way FCLoRA differs from AdaLoRA or MiLoRA, other than the forgetting motivation, for example by replacing your clipping rule with their allocation rule?

---

> ### Author Response · Authors · 2025-11-21
> **Authors Response**
>
> W1. Limited novelty compared with recent SVD based LoRA work.
>
> - We kindly refer the reviewer to the General Response section, 'Novelty of FCLoRA compard to SVD-based model'.
>
> W2. The crucial hyperparameter, the quantile of the pre-trained spectrum that defines the upper bound on Sigma, is under explained.
>
> -  We kindly refer the reviewer to the General Response section, 'Detailed explanation of $\bar{\sigma}$'.
>
> W3. The evidence for forgetting is weak. This is not the same as showing mitigation on a real continual learning sequence with several distinct downstream tasks and with standard CL metrics such as average accuracy and backward transfer.
>
> - We kindly refer the reviewer to the General Response section, 'Mitigation of Catastrophic Forgetting.'
>
> W4. Finally, the computational aspect of learning a parameterized SVD, including possible orthogonality regularization, is not quantified. There are no numbers for wall clock time, peak memory, or extra parameters, compared to standard LoRA or to another SVD style baseline.
>
> - In Appendix P, we have already reported the increase in parameter count, the analytical computational cost, and the comparison of training and inference time across LoRA, explicit SVD-based LoRA, and parameterized SVD–based LoRA variants. As demonstrated in these analyses, our method retains computational requirements that are fully comparable to the existing LoRA family.
>
> Q1. Can you run at least one sequential or continual experiment, for example a chain of four or five NLU tasks, and show that FCLoRA forgets slower than standard LoRA and at least one recent SVD LoRA baseline?
>
> - Continual learning with order: SST-2 -> CoLA -> RTE -> MRPC
>
> | Model | SST-2$_{finetune}$ | SST-2$_{pretrain}$ | CoLA$_{finetune}$ | CoLA$_{pretrain}$ | RTE$_{finetune}$ | RTE$_{pretrain}$ | MRPC$_{finetune}$ | MRPC$_{pretrain}$ |
> |--|--|--|---|--|---|--|--|----|
> | Pre-train | - | 61.64     | - | 61.64    | - | 61.64   | - | 61.64    |
> | LoRA | 94.50 | 45.15 | 59.74 | 48.51 | 71.00    | 40.47   | 88.97     | 33.38    |
> | MiLoRA    | 94.76 | 48.64 | 59.77 | 52.64    | 71.00    | 47.33   | 88.24     | 29.77    |
> | FCLoRA    | **95.37**  | **51.76** | **60.77** | **54.04**| **79.18**| **52.44**| **89.30** | **45.08**|
>
>
> Although our main paper focuses on single-step fine-tuning, we additionally investigate the effectiveness of our method in a continual learning setting where more than two tasks are learned sequentially. Specifically, we conduct four consecutive fine-tuning rounds on GLUE tasks. In this setup, the pre-trained weights are kept frozen, and only the LoRA weights are loaded from the previous task and further updated. We observe that our method exhibits a noticeably lower rate of forgetting compared to LoRA and MiLoRA.
>
> - Continual learning benchmark from [1], AA ($\uparrow$), BWT ($\uparrow$)
>
> | Model  | Order-1        | Order-2        | Order-3        | avg            |
> |---|--|----|----|---|
> | O-LoRA | 66.28 / -14.91  | 68.35 / -12.54 | 71.06 / -7.80   | 68.75 / -11.75 |
> | FCLoRA | **75.75 / -4.80** | **77.32 / -2.14** | **77.12 / -2.40** | **76.73 / -3.11** |
>
>
> To evaluate the performance of our model under a standard continual learning benchmark, we follow the experimental framework of O-LoRA [1], which is LoRA-based method for continual learning. For O-LoRA, we directly use the original implementation released on GitHub, and we measure both Average Accuracy (AA) and Backward Transfer (BWT). Following the original paper, AA is computed as the mean accuracy over all tasks when evaluated after completing the last task. For both metrics, higher is better.
>
> More detailed explanations and additional results are provided in the General Response 'Mitigation of Catastrophic Forgetting'.
>
>
> Q2. Can you provide a sensitivity analysis of the clipping quantile, for example per layer versus global, and quantiles?
>
> - We kindly refer the reviewer to the General Response section, 'Detailed explanation of $\bar{\sigma}$'.
>
>
> Q3. Can you report training time, peak memory and number of extra parameters for FCLoRA, for standard LoRA, and for one SVD LoRA baseline, on the same hardware and with the same sequence length?
>
> - As noted in our response to W4, Appendix P already reports the increase in parameter count, the analytical computational cost, and the comparison of training and inference time/GPU usage across LoRA, explicit SVD-based LoRA, and parameterized SVD-based LoRA variants.
>
>
> Q4. Can you clarify in a short ablation in which way FCLoRA differs from AdaLoRA or MiLoRA, other than the forgetting motivation, for example by replacing your clipping rule with their allocation rule?
>
> - We kindly refer the reviewer to the General Response, specifically '1. Ablation Study for Integrating Models' in 'Detailed explanation of $\bar{\sigma}$'.
>
> ---
>
> > [1] Wang, Xiao, et al. "Orthogonal subspace learning for language model continual learning." Findings of the Association for Computational Linguistics: EMNLP 2023. 2023.

---

> ### Comment · Reviewer_KCpv · 2025-11-26
>
> I appreciate the authors' effort in addressing the review comments. The additional continual learning experiments using the O-LoRA benchmark are valuable and do provide stronger evidence for the forgetting mitigation claims. The sensitivity analysis on σ̄ is also helpful for understanding the method's behavior.
>
> That said, I still have reservations about the core contribution. The authors argue that FCLoRA differs from prior work in motivation (forgetting mitigation rather than rank allocation or initialization), but from a technical standpoint, the method essentially comes down to applying clipping constraints on parameterized SVD. AdaLoRA already uses parameterized SVD with importance-based pruning, MiLoRA targets minor components, and PiSSA focuses on principal components for initialization. While the authors frame their approach from a different angle, the actual mechanism of constraining singular values during training does not appear fundamentally different from existing techniques, at least I did not find myself learning something genuinely new from the paper. It would be more convincing if the paper could articulate more clearly what fundamentally new insight or capability FCLoRA brings, beyond combining known ideas.
>
> Regarding the sensitivity analysis, I think it actually raises some concerns rather than fully resolving them. The results show that optimal quantile values vary across tasks (e.g., σ^(1) works best for some datasets while σ^(4) is better for others), which means practitioners would need to tune this hyperparameter carefully for each new application. This adds complexity and somewhat undermines the practical appeal of the method.
>
> The empirical improvements, while consistent, are relatively modest (typically in the range of 0.5-1.5% over baselines). Given the additional hyperparameter tuning required and the incremental nature of the technical contribution, I am not fully convinced that this meets the bar for ICLR.
>
> I am willing to raise my score slightly to acknowledge the authors' responsive rebuttal, but I maintain my overall assessment that the paper falls below the acceptance threshold.

---

> ### Author Response · Authors · 2025-11-27
>
> We sincerely appreciate your thoughtful assessment and thank you for raising your score in recognition of our rebuttal and additional experiments. Your careful feedback has been very helpful in clarifying  the contributions of this work.
>
> ---
>
> ### 1. Concern about FCLoRA’s contribution
>
> Contrary to the reviewer’s concern, FCLoRA proposes a direction that is distinct from existing LoRA-family approaches, even though it shares certain structural components such as SVD-based parameterization. In PEFT settings where computation and parameter budgets are limited, it is often more practical to employ methods that directly target the root cause of forgetting while maintaining computational efficiency, rather than adding architectural complexity. FCLoRA was designed based on this consideration: its contribution arises from a problem-driven formulation rather than structural simplicity.
>
> This work addresses two central challenges:
> (1) Injecting information into fine-grained minor components with small singular values to better utilize limited representational capacity
> (2) Structurally mitigating catastrophic forgetting through theoretically motivated constraints on the singular spectrum
>
> Existing methods primarily focus on rank allocation/ pruning, or component-based initialization. However, they do not include mechanisms to observe or regulate how minor components with small singular values evolve *during fine-tuning*. As a result, these components may unnecessarily inflate, leading to distortions in fine-grained representational dimensions and excessive singular value growth that weakens the pre-trained prior. These issues collectively contribute to catastrophic forgetting.
>
> FCLoRA explicitly identifies this overlooked problem and proposes an intervention strategy for stabilizing fine-grained singular components. Importantly, FCLoRA also yields consistent improvements even when combined with existing methods, without requiring additional hyperparameter tuning. If FCLoRA merely shared structural similarity with prior approaches, such independent improvements would not emerge. This provides empirical evidence that FCLoRA contributes at a different level by ensuring spectrum stability and fine-grained component control.
>
> Just as methods like LoRA+ [1] or rsLoRA [2] demonstrate that theoretically grounded yet simple design choices—such as using asymmetric learning rates for the A and B matrices, or scaling updates with $\alpha / \sqrt{r}$ —can yield gains, FCLoRA similarly shows that a conceptually simple but theoretically motivated intervention can have a significant practical impact. FCLoRA highlights the importance of singular spectrum control—an aspect largely overlooked in prior work. Evaluating FCLoRA solely based on structural simplicity may therefore underestimate its contribution.
>
> ---
> ### 2. Concern about tuning $\bar{\sigma}$
>
> While $\bar{\sigma}$ does affect performance, it should not be regarded as an overly burdensome hyperparameter. Experimental results show that fine-tuning performance remains relatively stable across a broad range of $\bar{\sigma}$ values, often matching or surpassing the baseline. In contrast, catastrophic forgetting varies substantially with $\bar{\sigma}$, as predicted by Theorem 3.1. This directly supports the underlying motivation of FCLoRA: $\bar{\sigma}$ governs the phenomenon that FCLoRA aims to address.
>
> Moreover, $\bar{\sigma}$ functions as a policy parameter controlling the degree of representational preservation and can be tuned as easily as commonly used hyperparameters such as learning rate, rank, or dropout. In practice, existing methods often require more complex tuning procedures, such as AdaLoRA’s multi-stage budget scheduling or the sensitive A/B learning rates in LoRA+. Considering that PEFT inherently involves hyperparameter-based optimization, tuning $\bar{\sigma}$ does not impose more burden than is typical for PEFT methods.
>
> ---
> ### 3. Claim that improvements are small
>
> Given the extremely limited parameter budget in LoRA-based fine-tuning, improvements of 0.5–1.5% are far from negligible. More importantly, FCLoRA provides substantial gains in mitigating catastrophic forgetting, beyond improvements in average task performance. For example, when fine-tuning RoBERTa on MRPC, the accuracy on the pre-trained task collapses to 3.77% under LoRA, whereas FCLoRA preserves it at 32%. This demonstrates that FCLoRA not only improves downstream performance but also retains pre-trained knowledge significantly better within the same parameter budget.
>
> We hope that the clarifications provided above help convey the distinct contribution of FCLoRA. We respectfully ask the reviewer to kindly reconsider the overall evaluation, and we are grateful once again for your time and feedback.
>
> ---
> > [1] Hayou et al. ,"LoRA+ efficient low rank adaptation of large models." ICML 2024
> >
> > [2] Kalajdzievski et al., "A rank stabilization scaling factor for fine-tuning with lora." preprint 2023

---

### Official Review · Reviewer_MCxD · 2025-10-30

**Soundness:** 2
**Presentation:** 3
**Contribution:** 2
**Rating:** 2
**Confidence:** 4

**Summary:**

This paper introduces FCLoRA, a variant of LoRA fine-tuning that injects "fine-grained" singular components into a frozen pre-trained model via a parametrized SVD. FCLoRA treats the adapter adjustment weights as $U\Sigma V^\top$ and clips each learnable singular value $\sigma_n$ to lie below a fixed threshold $\bar{\sigma}$, which is some q-th quantile of the original extracted singular values. As part of the objective function, the paper applies an orthogonality regularizer to $U$ and $V$. The main claim of the paper is that by focusing learning on smaller (or a range of) singular directions, FCLoRA adapts to new tasks while preserving the pre-trained structure, thus reducing catastrophic forgetting.

**Strengths:**

- A clear intuition that the dominant singular vectors of pre-trained weights should remain fixed while only finer components adapt. Then, the idea of parametrizing the adapter by SVD with slipping is a plausible way to realize this idea.
- Forgetting mitigation is demonstrably effective, though there is a drop in the pre-training accuracy.
- The authors do include an ablation table comparing variants of LoRA w./w.o. SVD parametrization, orthogonality constraint, and clipping. This is a good sign that the authors tried to analyze the components of their method.

**Weaknesses:**

- Several recent methods already (i) initialize/operate in SVD space, (ii) target minor components (milora), or (iii) decompose weight updates (dora). Here, the primary new element is clipping $\Sigma$ plus an interpretation of it as fine-grained injection. The paper should more rigorously contrast their clipping to: MiLoRA's only minor-component editing, AdaLoRA's rank budgeting, and LoRA^2's multi-scale SVD, with controlled ablations where only clipping differs.
- The theory is suggestive, but not decisive. The bound is loose and non-constructive wrt. an optimal threshold $\bar{\sigma}$; it motivates clipping but does not characterize when clipping is both necessary and sufficient. Spectral-norm growth study is cited to argue a typical increase in the norm, not the link to adapter-only updates, and optimizer types & batch size in these tasks are not quantified.
- Ablations are insufficient. Although Table 5 contrasts four variants, there are still mixed changes. The effects of each component (clipping threshold, orthogonalization, and SVD parametrization) are not cleanly disentangled. For example, there is no run with parametrized SVD without clipping, or clipping without orthogonalziation. This makes it harder to know which design choices actually drive the results.
- Note on the clipping threshold. The choice of $\bar{\sigma}$ is only briefly described as a fixed quantile of the pretrained singular values, and then used without justification. The main text offers no guidance on how it was chosen or how results would vary. Appendix O claims to study it, but those results & explanations need to be in the main paper. In practical usage, one would want to know if performance changes sharply or if fine-tuning on a certain set yields significantly different settings. Without this, a proper description of what is what, the method's robustness is unclear. It appears that, across models and tasks, this hyperparameter setting varies, adding another dimension of complexity.
- Forgetting is tested only on Bookcorpus with minimal detail on the setup, and just for text. The analysis lacks diversity across pre-training corpora or modalities.
- Despite "no overhead" claims, the appendix shows slower training due to orthogonal regularization. This contradicts the efficiency claim and should be reported clearly.
- **One other important note:** While the proposed spectral clipping intuitively restricts representational flexibility, the method consistently outperforms all baselines, including the unconstrained ones and across all datasets. This is counterintuitive. The paper should analyze why a capacity-limiting constraint leads to higher downstream accuracy. Is it because of uneven hyperparameter tuning or something else? For this reason, the empirical claims seem unconvincing.

**Questions:**

Please refer to the weaknesses.

---

> ### Author Response · Authors · 2025-11-21
> **Authors Response (1/3)**
>
> W1. The paper should more rigorously contrast their clipping to: MiLoRA's only minor-component editing, AdaLoRA's rank budgeting, and LoRA$^2$'s multi-scale SVD, with controlled ablations where only clipping differs.
>
> A1. Please refer to our general response 'Novelty of FCLoRA compard to SVD-based model' and 'Detailed explanation of $\bar{\sigma}$'.
>
>
> W2. The theory is suggestive, but not decisive. The bound is loose and non-constructive wrt. an optimal threshold $\bar{\sigma}$; it motivates clipping but does not characterize when clipping is both necessary and sufficient. Spectral-norm growth study is cited to argue a typical increase in the norm, not the link to adapter-only updates, and optimizer types & batch size in these tasks are not quantified.
>
> A2.
>
> > the bound is suggestive but not decisive, and that it is loose or non-constructive
>
> The theoretical section is not intended to fully characterize the MAP posterior. Its purpose is to clarify a structural relationship: as the singular values of the adapter grow, the prior probability decreases. The theorem shows that large singular-value growth provides a mechanism through which the pre-trained prior can be degraded.
>
> Accordingly, the role of the bound is not to compute an optimal threshold. Rather, it provides the theoretical rationale for why spectral clipping naturally emerges as a design choice. The threshold $\bar{\sigma}$ is not determined directly by the theory; instead, it is set using statistics of the pre-trained spectrum (e.g., quartiles), and its sensitivity is systematically evaluated in the appendix. To address the reviewer’s concern, we have added more detailed explanation of this sensitivity analysis in the General Response.
>
>
> > The theory motivates clipping but does not characterize when it is necessary or sufficient
>
> The paper does not claim that clipping is necessary or sufficient. It makes two concrete points:
>
> 1. When singular values grow without constraint, the pre-trained prior inevitably deteriorates, which aligns with the regimes where catastrophic forgetting is observed (see Figure 1 (b), Figure 3).
> 2. Introducing a mechanism that keeps the singular values within a reasonable range prevents entry into this regime, and empirically this substantially mitigates forgetting.
>
> A complete necessary-and-sufficient characterization would require strong assumptions about the task distribution, data size, and optimization dynamics. The paper intentionally avoids such assumptions and instead isolates ‘large singular values’ as a risk factor for prior degradation, and proposes clipping as a practical mechanism that directly targets this factor.
>
>
> > Regarding spectral-norm growth, adapter-only updates, and specification of optimizer and batch size
>
> The cited spectral-norm growth results establish that norm growth is a typical phenomenon in stochastic optimization. The paper extends this connection to the adapter-only setting.
>
> - In LoRA-based methods, $\theta - \theta_0 = \Delta W$, so the spectrum discussed in the theory corresponds exactly to the spectrum of the adapter.
> - Figure 3(a) tracks the spectral norm of $\Delta W$ directly. LoRA and its variants show steady norm growth during tuning, while FCLoRA keeps this norm controlled through clipping. This links the general theoretical trend to the concrete dynamics of adapter-only updates.
>
> Regarding optimizer and batch size, all experiments use standard AdamW-type optimizers and learning-rate/batch-size ranges consistent with established fine-tuning setups. These details are summarized in the experimental appendix.

---

> ### Author Response · Authors · 2025-11-21
> **Authors Response (2/3)**
>
> W3. Ablations are insufficient. The effects of each component are not cleanly disentangled. For example, there is no run with parametrized SVD without clipping, or clipping without orthogonalziation.
>
> A3.
> In Table 5 of the main text, we conducted an ablation study on the injected components. Among these, the setting referred to by the reviewer as “run with parameterized SVD without clipping” corresponds to 'LoRA$_{SVD}$'.
> To address the reviewer’s concern, we additionally performed an ablation study in which clipping is applied without orthogonal regularization, and we expanded the experimental results accordingly. However, because this setting does not ensure the orthonormality of the singular vectors, Theorem 3.1 cannot be applied, and thus it lacks a theoretical guarantee that catastrophic forgetting is consistently mitigated (Table 5).
>
> | Model              | SST-2$_{finetune}$ | SST-2$_{pretrain}$ | RTE$_{finetune}$ | RTE$_{pretrain}$ | MRPC$_{finetune}$ | MRPC$_{pretrain}$ |
> |--------------------|------------|-----------|----------|---------|-----------|----------|
> | Pre-trained        | -          | 61.64     | -        | 61.64   | -         | 61.64    |
> | LoRA               | 94.80      | 32.35     | 80.39    | 33.78   | 89.05     | 3.77     |
> | LoRA_UV            | 94.75      | 39.39     | 81.47    | 38.01   | 89.22     | 3.12     |
> | LoRA_SVD           | 95.03      | 49.65     | 82.55    | 40.65   | 89.62     | 17.57    |
> | LoRA_clip&no ortho | 94.84      | 45.74     | 82.31    | 51.75   | 90.20     | 28.31    |
> | FCLoRA             | **95.37**  | **51.29** | **83.15**| **50.07**| **90.32** | **32.00**|
>
>
> Furthermore, in the General Response 'Detailed explanation of $\bar{\sigma}$', we conducted several ablation and sensitivity studies regarding $\bar{\sigma}$, including:
>
> - Ablation study on learnable $\bar{\sigma}$
> - Comparison between global and layer-wise quantile settings
> - Sensitivity analysis of the quantile on performance and the mitigation of catastrophic forgetting
>
> W4. Note on the clipping threshold. The choice of is only briefly described as a fixed quantile of the pretrained singular values, and then used without justification. The main text offers no guidance on how it was chosen or how results would vary. Appendix O claims to study it, but those results & explanations need to be in the main paper. In practical usage, one would want to know if performance changes sharply or if fine-tuning on a certain set yields significantly different settings. Without this, a proper description of what is what, the method's robustness is unclear. It appears that, across models and tasks, this hyperparameter setting varies, adding another dimension of complexity.
>
> We will incorporate the contents of Appendix O into the main paper to enhance the clarity and completeness of our presentation. In addition, to address the reviewer’s concerns, we have conducted further experiments and expanded our empirical analysis accordingly. We kindly refer the reviewer to the General Response 'Detailed explanation of $\bar{\sigma}$' for detailed explanations and additional results.
>
>
> W5. Forgetting is tested only on Bookcorpus with minimal detail on the setup, and just for text. The analysis lacks diversity across pre-training corpora or modalities.
>
> - We have already reported the forgetting results on additional tasks and models in Appendix N, 'Further Experiments on Mitigating Catastrophic Forgetting'. To further address the reviewer’s concerns, we conducted a more extensive set of additional experiments and expanded our empirical analysis. For comprehensive explanations and the full set of supplementary results, we kindly refer the reviewer to the General Response section, 'Mitigation of Catastrophic Forgetting.'

---

> ### Author Response · Authors · 2025-11-21
> **Authors Response (3/3)**
>
> W6. Despite "no overhead" claims, the appendix shows slower training due to orthogonal regularization.
>
> We apologize for any confusion caused. Our statement regarding 'no overhead' refers specifically to the inference stage after fine-tuning, as clarified in the main text. Since orthogonal regularization is not used once training is completed, it introduces no additional overhead during inference. Furthermore, our training time is comparable to—or even lower than—that of strong parameterized SVD-based baselines such as AdaLoRA, which we believe is reasonable given the performance and the method’s ability to mitigate catastrophic forgetting.
>
>
> W7. While the proposed spectral clipping intuitively restricts representational flexibility, the method consistently outperforms all baselines, which is counterintuitive. The paper should analyze why a capacity-limiting constraint leads to higher downstream accuracy.
>
> [FCLoRA focus capacity on useful representations rather than restricting it]
>
> The strength of a pre-trained LM largely stems from its ability to encode global world knowledge learned from a massive corpus. In many open-source LLMs, the pre-training stage is also understood as the phase during which the model acquires broad, transferable knowledge useful for downstream tasks. Therefore, the objective of fine-tuning—particularly in PEFT—is not to “re-learn” new knowledge, but rather to reuse and adapt this pre-trained global knowledge efficiently within a limited parameter budget.
>
> As illustrated in Figure 1, principal components associated with large singular values exhibit high Fisher overlap between the pre-training and fine-tuning tasks and are thus already well aligned. In contrast, components associated with small singular values tend to be fine-grained and task-specific, requiring greater adaptation. From this perspective, the spectral clipping in FCLoRA should not be viewed as a constraint that reduces overall capacity, but rather as a mechanism that allocates the limited parameter budget more effectively between the aligned (global) and task-specific (fine-grained) components.
>
> Concretely, FCLoRA preserves the dominant components to maintain pre-trained knowledge, while injecting fine-grained information only within a controlled, clipped singular value range. This encourages the model to selectively learn new directions (unaligned singular vectors) absent in the pre-trained weight. Such clipping acts as a regularizer that prevents the representation space from expanding excessively in undesirable directions and, from a MAP viewpoint, helps preserve the pre-training prior, thereby reducing catastrophic forgetting.
>
> As shown in Appendix L, FCLoRA not only acquires fine-grained information absent from the pre-trained weights but also maintains pre-trained task performance more effectively than existing LoRA variants. This supports that the performance gains arise not from hyperparameter variance, but from the proposed spectral constraint’s ability to focus capacity on useful representations rather than restricting it.
>
> [Singular Value Clipping (SVP) in other domains]
>
> Moreover, singular value clipping (SVP), which is closely related to our approach, has been widely used across diverse domains such as GANs and CNNs. For example, [1] introduces a K-Lipschitz constraint to overcome instability in GANs and achieves it via SVP. In [2], the authors show that the singular values of convolutional layers are key contributors to exploding and vanishing gradients; by controlling the Lipschitz constant through SVP, they prevent these issues and improve generalization performance. Additionally, appropriate clipping has been shown to enhance network stability and test-time generalization. Similarly, [3] demonstrates that regularizing per-layer spectral norms improves both model generalization and adversarial robustness.
>
> [Conclusion]
>
> Therefore, our method does not hinder the model’s expressivity; rather, it offers LoRA-specific theoretical justification—such as rank-group task alignment and mitigation of catastrophic forgetting—while also acting in a manner consistent with prior findings across various domains that SVP can enhance training stability and improve generalization. This makes our approach a complementary and strengthening component for LoRA-based fine-tuning.
>
>
> > [1] Saito, Masaki, Eiichi Matsumoto, and Shunta Saito. "Temporal generative adversarial nets with singular value clipping." Proceedings of the IEEE international conference on computer vision. 2017.
>
> > [2] Senderovich, Alexandra, et al. "Towards practical control of singular values of convolutional layers." Advances in Neural Information Processing Systems 35 (2022): 10918-10930.
>
> > [3] Boroojeny, Ali Ebrahimpour, Matus Telgarsky, and Hari Sundaram. "Spectrum extraction and clipping for implicitly linear layers." International Conference on Artificial Intelligence and Statistics. PMLR, 2024.

---

### Author Response · Authors · 2025-11-21
**General Response #3 -  Mitigation of catastrophic forgetting**

Since Theorem 3.1 characterizes how the growth of singular values leads to forgetting, it shows that FCLoRA can mitigate catastrophic forgetting. In addition to the analyses in Figure 3 and Table 5, we have already conducted further experiments on catastrophic forgetting for RoBERTa and LLaMA in Appendix N, titled 'Further experiments on mitigating catastrophic forgetting'.

To further address the reviewers’ concerns, we additionally conducted more experiments on catastrophic forgetting. Note that for DeBERTa, issues in the publicly released pre-trained weights related to the pre-training head prevent running this evaluation. For the C4-en dataset, we sampled the top 10k test examples for measurement.

1. RoBERTa + OpenWebText, Accuracy ($\uparrow$)

| Model       | SST2$_{finetune}$ | SST2$_{pretrain}$ | CoLA$_{finetune}$ | CoLA$_{pretrain}$ | QNLI$_{finetune}$ | QNLI$_{pretrain}$ | RTE$_{finetune}$ | RTE$_{pretrain}$ | MRPC$_{finetune}$ | MRP$_{pretrain}$ |
|-|--|-|--|-|--|--|-|--|-|--|
| Pre-Trained | -         | 68.21    | -         | 68.21    | -         | 68.21    | -        | 68.21    | -          | 68.21     |
| LoRA        | 94.8      | 54.33    | 64.49     | 63.67    | 92.73     | 62.88    | 80.39    | 53.47    | 89.05      | 18.36     |
| FCLoRA      | **95.37** | **65.67**| **64.79** | **64.31**| **93.09** | **63.59**| **83.15**| **62.69**| **90.32**  | **47.27** |


2. RoBERTa + STORIES, Accuracy ($\uparrow$)


| Model       | SST2$_{finetune}$ | SST2$_{pretrain}$ | CoLA$_{finetune}$ | CoLA$_{pretrain}$ | QNLI$_{finetune}$ | QNLI$_{pretrain}$ | RTE$_{finetune}$ | RTE$_{pretrain}$ | MRPC$_{finetune}$ | MRP$_{pretrain}$ |
|--|---|--|--|--|--|--|---|---|---|---|
| Pre-Trained | -         | 69.49| -         | 69.49| -         | 69.49| -        | 69.49| -          | 69.49 |
| LoRA        | 94.8      | 40.13    | 64.49     | 61.87    | 92.73     | 62.89    | 80.39    | 45.90    | 89.05      | 8.42      |
| FCLoRA      | **95.37** | **60.38**    | **64.79** | **64.45**| **93.09** | **63.91**| **83.15**| **60.31**    | **90.32**  | **41.12** |


3. LLaMA + PG19, Accuracy ($\uparrow$) / PPL ($\downarrow$)

| Model       | LLaMA-7B Acc$_{finetune}$ | LLaMA-7B PPL$_{pretrain}$ | LLaMA-2-7B Acc$_{finetune}$ | LLaMA-2-7B PPL$_{pretrain}$ |
|---|---|-|---|-------|
| Pre-Trained | - | 6.66 | - | 6.61 |
| LoRA        | 74.7 | 8.69 | 77.6 | 11.36 |
| FCLoRA      | **78.6** | **7.78** | **81.3** | **8.92** |

4. LLaMA + C4-en, Accuracy ($\uparrow$) / PPL ($\downarrow$)

| Model       | LLaMA-7B Acc$_{finetune}$ | LLaMA-7B PPL$_{pretrain}$ | LLaMA-2-7B Acc$_{finetune}$ | LLaMA-2-7B PPL$_{pretrain}$ |
|---|--|----|----|---|
| Pre-Trained | - | 6.66 | - | 6.61 |
| LoRA        | 74.7 | 8.69 | 77.6 | 11.36 |
| FCLoRA      | **78.6** | **7.78** | **81.3** | **8.92** |


- Continual Learning on GLUEwith order: SST-2 -> CoLA -> RTE -> MRPC


| Model     | SST-2$_{finetune}$ | SST-2$_{pretrain}$ | CoLA$_{finetune}$ | CoLA$_{pretrain}$ | RTE$_{finetune}$ | RTE$_{pretrain}$ | MRPC$_{finetune}$ | MRPC$_{pretrain}$ |
|-----------|------------|-----------|-----------|----------|----------|---------|-----------|----------|
| Pre-train | -          | 61.64     | -         | 61.64    | -        | 61.64   | -         | 61.64    |
| LoRA      | 94.50      | 45.15     | 59.74     | 48.51    | 71.00    | 40.47   | 88.97     | 33.38    |
| MiLoRA    | 94.76      | 48.64     | 59.77     | 52.64    | 71.00    | 47.33   | 88.24     | 29.77    |
| FCLoRA    | **95.37**  | **51.76** | **60.77** | **54.04**| **79.18**| **52.44**| **89.30** | **45.08**|

- Although our main paper focuses on a single round of fine-tuning, we also evaluated FCLoRA in a continual learning setting where multiple tasks are learned sequentially. Specifically, we trained four GLUE tasks in sequence. During this process, we kept the pre-trained weights frozen and loaded only the LoRA weights from the previous task before training the next one. Compared to LoRA and MiLoRA, FCLoRA exhibited significantly slower forgetting.


- Continual learning on Benchmark Task

| Model  | Order-1        | Order-2        | Order-3        | avg            |
|--------|-----------------|----------------|-----------------|----------------|
| O-LoRA | 66.28 / -14.91  | 68.35 / -12.54 | 71.06 / -7.80   | 68.75 / -11.75 |
| FCLoRA | **75.75 / -4.80** | **77.32 / -2.14** | **77.12 / -2.40** | **76.73 / -3.11** |

 To evaluate our model in a benchmark continual learning setting, we followed the experimental framework of O-LoRA [1]. We used the publicly released original code from the O-LoRA repository and measured both Average Accuracy (AA) and Backward Transfer (BWT). Following the original paper, AA was computed as the average accuracy over all tasks after completing the final task.


> [1] Wang, Xiao, et al. "Orthogonal subspace learning for language model continual learning." Findings of the Association for Computational Linguistics: EMNLP 2023. 2023.

---

### Author Response · Authors · 2025-11-21
**General Response #2 -  Detailed explanation of $\bar{\sigma}$**

Setting $\bar{\sigma}$ as a quantile of the singular values of the pre-trained weights is a natural design choice, as it inherently accounts for the spectrum distribution across different models and layers. In practice, the spectrum distribution varies depending on the model size and the type of layer. For example, both across different models and within the same model, the largest singular value of the pre-trained weights can differ significantly.

- In LLaMA-7B, the first query layer has a largest singular value of 32.21, whereas in RoBERTa the first query layer has 10.20.
- Within the same model, the first query layer of LLaMA-7B has a singular value of 32.21, while the last query layer has 15.49.
- Within the same model, the first query layer of LLaMA-7B has 32.21, whereas the first value layer has 4.38.

Thus, assigning a global constant value is not suitable for scalability across models. Moreover, since fine-grained components should consider their relative position within each singular spectrum, our quantile-based design is appropriate. Naturally, the choice of quantile varies depending on the task or model and can be configured as a hyperparameter.

However, since we understand the reviewers’ concerns, we conducted several experiments regarding $\bar{\sigma}$.


1. Ablation study on learnable $\bar{\sigma}$

| Model           | CoLA     | RTE      | MRPC     | STS-B     |
|-----------------|----------|----------|----------|-----------|
| Learnable sigma | 64.21    | 82.43    | 90.28    | 91.10     |
| FCLoRA          | **64.79**| **83.15**| **90.32**| **91.22** |

This experiment evaluates the performance when sigma_bar is learnable. We observed that using a fixed constant yields better performance than allowing sigma_bar to be learnable.


2. Ablation study on global or layer-wise quantile


| RoBERTa   | CoLA     | RTE      | MRPC     | STS-B     |
|-----|----|------|----|--|
| Descending ($\sigma^{(4)}\rightarrow\sigma^{(1)}$)   | 63.03    | 83.03    | 89.87    | 90.84     |
| Ascending ($\sigma^{(1)}\rightarrow\sigma^{(4)}$)    | 63.40    | 80.99    | 88.64    | 91.06     |
| FCLoRA (Global)     | **64.79**| **83.15**| **90.32**| **91.22** |

While the original FCLoRA uses a uniform quantile across all layers, we evaluated the effect of varying the quantile on a layer-wise basis. In the descending setting, the quantile decreases as the layer index increases, dividing all layers into four groups. For example, in a 32-layer architecture with 'Descending', the top eight layers use $\bar{\sigma}=\sigma^{(4)}$, the next eight layers use $\bar{\sigma}=\sigma^{(3)}$, and so on. The ascending setting applies the reverse order. Although the optimal strategy may vary across tasks, using a single consistent quantile across all layers consistently yielded the most stable and strong performance.


3. Sensitivity study for quantile on performance and mitigation of catastrophic forgetting:

| Task | Metric        | 0.1     | 0.3     | 0.5     | 0.7     | 0.9     |
|------|---------------|---------|---------|---------|---------|---------|
| CoLA | Acc$_{finetune}$  | 64.17   | 64.60  | **64.79** | 63.32  | 62.59  |
|      | Acc$_{pretrain}$  | 42.76   | 47.03  | **54.78** | 50.47  | 52.28  |
| MRPC | Acc$_{finetune}$  | 89.95   | 89.71  | **90.32** | 89.95  | 89.62  |
|      | Acc$_{pretrain}$  | 7.41    | 25.80  | 32.00  | 31.21  | **42.25** |
| STSB | Acc$_{finetune}$  | **91.09** | 91.04  | 91.08  | 91.02  | 91.02  |
|      | Acc$_{pretrain}$  | 47.51   | **51.34** | 45.99  | 50.92  | 51.29  |
| RTE  | Acc$_{finetune}$  | 80.14   | 80.99  | 81.71  | 81.95  | **83.03** |
|      | Acc$_{pretrain}$  | 28.29   | 41.25  | 39.50  | **48.45** | 45.57  |

| Model       | 0.1            | 0.3            | 0.5            | 0.7            | 0.9            |
|-------------|----------------|----------------|----------------|----------------|----------------|
| SQuAD1 $r=4$  | **88.1**/93.8  | 87.8/93.8  | **88.1**/**93.9**      | 88.0/**93.9**      | 88.0/93.8      |
| SQuAD1 $r=8$  | 88.0/94.1  | 87.8/93.8  | **88.6**/**94.3** | 88.0/93.9      | 87.9/93.8      |



We measured the quantile-dependent performance variation for NLU and QA tasks, and additionally evaluated sensitivity to catastrophic forgetting for NLU tasks. (For DeBERTa, issues in the publicly released model related to the pre-training head prevented us from conducting experiments to directly measure pre-training performance. [link](https://github.com/microsoft/DeBERTa/issues/74)) The results show that when the quantile is too small and $\bar{\sigma}$ becomes excessively large, pre-training performance degrades significantly. This effect is particularly strong for MRPC and RTE when the quantile is 0.1, where performance drops sharply. In contrast, when the quantile increases to an appropriate range and sigma_bar becomes sufficiently small, we find that both pre-training retention and downstream fine-tuning performance can be simultaneously achieved.

---

### Author Response · Authors · 2025-11-21
**General Response #1 - Novelty of FCLoRA compared to SVD-based model**

Our methodology is an efficient approach rigorously devised from empirical evidence for adaptation across the singular value groups of the pre-trained weight (Figure 1) and the theoretical foundation for forgetting (Theorem 3.1). We explained the works referenced by the reviewers (PiSSA, MiLoRA, AdaLoRA) in Related Work Section 2.2, included SVD-based models as baselines in Table 1 to Table 3 of the main text to empirically demonstrate the effectiveness of our method in adaptation, and further showed in Section 5.1 that our approach addresses the limitations of Explicit SVD-based LoRA. In Appendix N.2, we also demonstrated that our method mitigates forgetting compared to parameterized SVD-based LoRA (AdaLoRA, SorSA).


Moreover, contrary to the reviewers' statement, the use of parameterized SVD in AdaLoRA and LoRA$^2$ was intended for rank allocation rather than singular value analysis. SorSA applies parameterized SVD on top of PiSSA's initialization scheme. Therefore, none of these methods address the magnitude of singular values *during* fine-tuning. Spectral initialization methods, including PiSSA and MiLoRA, only propose initializing adapters based on standard SVD theory and do not consider the evolution of the spectrum throughout fine-tuning. Consequently, as shown in Figure 3, the spectral norm of the adapter grows rapidly during fine-tuning. We have explicitly discussed all such points in the related work and the appendix.


Nevertheless, considering the remaining concerns raised by the reviewers, we conducted additional experiments.

1. Ablation study for integrating FCLoRA with existing models

| Method            |  CoLA | QNLI  | RTE   | MRPC  | STS-B |
|-------------------|-------|-------|-------|-------|-------|
| LoRA              | 64.49 | 92.73 | 80.39 | 89.05 | 90.87 |
| MiLoRA            | 64.31 | 92.96 | **81.35** | **89.30** | 90.06 |
| MiLoRA + FCLoRA   | **64.57** | **93.11** | 81.23 | **89.30** | **91.00** |
| AdaLoRA           | 61.37 | 92.54 | 81.11 | 89.05 | 90.62 |
| AdaLoRA + FCLoRA  | **64.30** | **93.16** | **82.67** | **89.78** | **91.02** |
| FCLoRA            | **64.79** | 93.09 | **83.15** | **90.32** | **91.22** |

Our method generates fine-grained components by clipping singular values according to the spectrum of the pre-trained weights under a fixed rank. The proposed approach can be integrated with models using parameterized SVD such as AdaLoRA and LoRA$^2$, as well as spectral initialization methods such as PiSSA and MiLoRA. We evaluated the performance by integrating our method with promising approaches like AdaLoRA and MiLoRA. In this ablation, both the backbone models and FCLoRA used the best hyperparameters identified for each model. As shown in the table below, integrating our method leads to substantial performance improvements in most cases. This indicates that existing models still have room to utilize their limited parameters more effectively and that learning fine-grained components contributes significantly to improving adaptation.

2. More comparisons on baselines

| Model       | PEFT   | BoolQ | PIQA | SIQA | HellaSwag | WinoGrande | ARC-e | ARC-c | OBQA | Avg. |
|-------------|--------|-------|------|------|-----------|------------|-------|-------|------|------|
| ChatGPT     | -      | 73.1  | 85.4 | 68.5 | 78.5      | 66.1       | 89.8  | 79.9  | 74.8 | 77.0 |
| LLaMA2-7B   | LoRA   | 69.8  | 79.9 | 79.5 | 83.6      | 82.6       | 79.8  | 64.7  | 81.0 | 77.6 |
|             | PiSSA  | 67.6  | 78.1 | 78.4 | 76.6      | 78.0       | 75.8  | 60.2  | 75.6 | 73.8 |
|             | MiLoRA | 67.6  | 83.8 | 80.1 | 88.2      | 82.0       | 82.8  | 68.8  | 80.6 | 79.2 |
|             | FCLoRA | 73.2  | 82.9 | 79.8 | 91.9      | 83.0       | 85.2  | 71.6  | 82.6 | **81.3** |

| QA      | SQuADv1.1 (0.08%) | SQuADv1.1 (0.65%) |
|---------|-------------------|-------------------|
| AdaLoRA | 87.2/93.4         | 87.6/93.7         |
| MiLoRA  | 87.1/93.3         | 85.5/92.0         |
| PiSSA   | 87.2/93.2         | 85.6/92.1         |
| FCLoRA  | **88.1/93.9**         | **88.9/94.3**         |


We additionally report results for commonsense reasoning and question answering to further validate the performance of FCLoRA. For commonsense reasoning, the PiSSA and MiLoRA results are taken from the MiLoRA paper, and for QA we tuned the models using the learning rate configured for AdaLoRA. The results confirm that our method continues to outperform baselines across both tasks.

---

### Note · Authors · 2026-01-05

I have read and agree with the venue's withdrawal policy on behalf of myself and my co-authors.